# HEPHAESTUS: Hierarchical Periodic Heterogeneous Adaptive Spatio-Temporal Unified System for Traffic Forecasting

## Abstract

Accurate traffic forecasting requires modeling complex spatio-temporal dynamics characterized by multi-scale temporal patterns, periodic dependencies, and spatial heterogeneity. While recent advances in spatio-temporal graph neural networks (STGNNs) have improved predictive performance, they often rely on fixed architectures that lack adaptivity to input-driven variations in temporal granularity and spatial connectivity. In this work, we propose **HEPHASTUS**[1], a novel framework that unifies adaptive multi-scale temporal modeling, explicit periodicity-aware attention, and dynamic spatial heterogeneity learning within a lightweight, scalable architecture. Our approach introduces three key components: (i) an Adaptive Multi-Scale Mixture of Experts (AMS-MoE) that deeply integrates multi-scale modeling with expert routing, using a dynamic router to automatically assign input sequences of different time scales to specialized experts, each expert focuses on temporal feature extraction at a specific scale (e.g., local fluctuations or long-term trends), with scale weights adaptively adjusted according to the time-varying input characteristics, enabling collaborative capture of global dependencies and local details; (ii) a Periodic Temporal Attention (PTA) mechanism that explicitly captures daily and weekly patterns via parameterized period matrices; and (iii) a Heterogeneous Spatial Attention (HSA) module that balances global structure and local specificity through node embeddings and a learnable pattern library, with low parametric cost. Experiments on six real-world traffic datasets,PeMS-BAY, METR-LA, PEMS03 and others, show that the proposed method achieves state-of-the-art performance across MAE, RMSE, and MAPE, with consistent gains over existing baselines. Ablation studies confirm the necessity of each design choice. Our results highlight the importance of adaptive, structured modeling in capturing the intrinsic dynamics of urban traffic.

## 1 Introduction

Accurate traffic forecasting is a cornerstone of intelligent transportation systems (ITS), enabling proactive traffic management, route guidance, and urban planning. However, the task remains highly challenging due to the complex spatio-temporal dynamics in traffic networks—characterized by intricate spatial dependencies among road segments and multi-granularity temporal patterns across various timescales (e.g., short-term fluctuations, daily cycles, and weekly trends). Moreover, traffic patterns exhibit significant spatial heterogeneity and evolve dynamically over time, further complicating accurate modeling.

Over the past decade, Spatio-Temporal Graph Neural Networks (STGNNs) have become the dominant paradigm for traffic prediction, effectively combining graph-based spatial modeling with sequential temporal learning. Early methods either treated traffic data as independent time series (Drucker et al., 1996; Ariyo et al., 2014) or relied on fixed graph structures to model spatial relationships (Yu et al., 2018; Li et al., 2018). While foundational, these approaches often fail to

---

[1]In Greek mythology, Hephaestus was renowned for his ability to create intricate tools and structures through adaptive craftsmanship—a fitting analogy for our model's multi-scale adaptive routing and heterogeneous fusion mechanisms.

capture the dynamic and heterogeneous nature of real-world traffic systems. Subsequent models incorporated attention mechanisms to disentangle periodic dependencies (Guo et al., 2019), integrate node identity information (Shao et al., 2022a; Liu et al., 2023a), or leveraged pre-training strategies for long-term pattern extraction (Shao et al., 2022b; Han et al., 2024; Gao et al., 2024). Despite these advances, most existing methods employ rigid architectural designs that struggle to adapt to varying temporal granularities and spatial configurations in real traffic data.

A critical limitation lies in modeling multi-scale temporal dynamics. Although recent works have explored hierarchical or multi-resolution architectures (Liu et al., 2022; Shabani et al., 2023; Wu et al., 2023; Wang et al., 2024b), they typically adopt fixed scale decomposition strategies that cannot flexibly respond to the intrinsic dynamics of input sequences. This rigidity hinders their ability to simultaneously capture transient shocks and persistent periodic trends. In the spatial domain, many models rely on static adjacency matrices based on geographic distance, which fail to reflect time-varying connectivity and functional similarities among sensors. While adaptive (Wu et al., 2019) and dynamic graph learning approaches (Li et al., 2023; Shin & Yoon, 2024) have been proposed, they often incur high parameter costs or fail to balance global consistency with local specificity in large-scale networks.

To address these challenges, we propose **HEPHASTUS**, a novel spatio-temporal forecasting framework that unifies adaptive multi-scale temporal modeling, periodicity-aware attention, and dynamic spatial heterogeneity learning within an efficient and scalable architecture. Our approach introduces three key innovations. First, we design an **Adaptive Multi-Scale Mixture of Experts (AMS-MoE)** module that enables input-dependent decomposition and fusion of temporal patterns across multiple scales via a differentiable routing mechanism—moving beyond fixed-scale modeling. Second, we propose a **Periodic Temporal Attention (PTA)** mechanism that explicitly models daily and weekly periodicities through learnable period matrices $P_D$ and $P_W$, enhancing long-term dependency capture via spatio-temporal decoupled computation. Third, we develop a **Heterogeneous Spatial Attention (HSA)** mechanism grounded in node embeddings $S_e$ and a learnable pattern library $PL$, achieving a dynamic trade-off between shared global structures and node-specific behaviors with low parametric overhead ($O(Nr + rCD)$).

Extensive experiments on six real-world traffic datasets—PeMS-BAY, METR-LA, PEMS03, PEMS04, PEMS07, and PEMS08—demonstrate that HEPHASTUS achieves state-of-the-art (SOTA) performance across major evaluation metrics, including MAE, RMSE, and MAPE, consistently outperforming existing baselines. Ablation studies validate the effectiveness and necessity of each component. In summary, this work advances spatio-temporal modeling in traffic forecasting by introducing a unified, adaptive, and efficient framework that better aligns with the multi-scale, periodic, and heterogeneous nature of real-world traffic systems.

## 2 RELATED WORK

**Spatio-Temporal Graph Neural Networks (STGNNs).** Early traffic forecasting methods (Drucker et al., 1996; Ariyo et al., 2014) relied on classical time series models but failed to capture spatial dependencies. With the rise of graph neural networks, Yu et al. (2018) pioneered a purely convolutional architecture for spatio-temporal graph modeling, establishing a foundational framework. Li et al. (2018) modeled traffic flow as a diffusion process on directed graphs and introduced diffusion graph convolution to better capture spatial dynamics. To address periodic patterns, Guo et al. (2019) decomposed temporal dependencies into recent, daily, and weekly components, using attention mechanisms to model them separately. More recently, Shao et al. (2022a); Liu et al. (2023a) proposed incorporating spatial and temporal identity embeddings to resolve sample indistinguishability without relying on complex graph structures. Concurrently, pre-training frameworks such as those in Shao et al. (2022b); Han et al. (2024); Gao et al. (2024) have shown effectiveness in capturing long-term spatio-temporal patterns from historical data, which are then transferred or integrated into downstream forecasting models.

**Multi-Scale Temporal Modeling.** Multi-scale and hierarchical architectures have demonstrated success across domains including computer vision, NLP, and time series analysis. By capturing features at varying temporal granularities, these models can better represent both transient fluctuations and persistent trends. Liu et al. (2022) proposed a pyramid attention mechanism to extract hierarchical temporal representations. Shabani et al. (2023) progressively refines predictions across

scales and introduces cross-scale normalization to mitigate distribution shifts. Wu et al. (2023) transforms 1D time series into 2D tensors via Fourier-based reparameterization, enabling 2D convolutions to model intra- and inter-scale dynamics. Wang et al. (2024b;a) decompose time series into multi-scale components and mix seasonal and trend patterns in both fine-to-coarse and coarse-to-fine manners, effectively aggregating microscopic and macroscopic information. However, most existing approaches employ fixed scale decomposition strategies and lack the ability to adaptively adjust the modeling hierarchy based on input dynamics—a limitation our work aims to overcome.

**Dynamic Spatial Modeling.** Conventional GNN-based methods often rely on static adjacency matrices (e.g., distance-based graphs) to define the topology of traffic networks. However, real-world multivariate time series typically lack explicit graph structures, necessitating data-driven discovery of node relationships. Moreover, traffic network topologies exhibit significant temporal variability due to changing traffic conditions, rendering static graphs insufficient. To this end, Wu et al. (2019) introduced a learnable adaptive adjacency matrix, allowing the model to infer latent spatial dependencies from data. Li et al. (2023) further advanced this idea by dynamically generating adjacency matrices at each time step, significantly improving adaptability in complex urban environments. Shin & Yoon (2024) constructed progressive graphs by learning trend similarities between nodes, combined with diffusion causal convolution and gated activation units for enhanced temporal modeling. While effective, such methods often incur high parameter overhead or lack explicit mechanisms to balance global consistency with local specificity—challenges addressed in our approach.

## 3 Problem Definition

Given a historical traffic sequence $X_{t-H+1:t} \in \mathbb{R}^{N \times H \times d}$ over the past $H$ time steps, the goal of traffic forecasting is to predict the future $F$ time steps by learning a model $\mathcal{M}(\cdot)$ with parameters $\theta$. Formally, this can be expressed as:

$$[X_{t-H+1}, \ldots, X_t] \xrightarrow[\theta]{\mathcal{M}(\cdot)} [X_{t+1}, \ldots, X_{t+F}], \tag{1}$$

where each observation $X_i \in \mathbb{R}^{N \times d}$, $N$ denotes the number of spatial nodes (e.g., sensors), and $d$ is the feature dimension. In our setting, $d = 1$, representing either traffic volume or speed.

## 4 Methodology

### 4.1 Model Architecture

The overall architecture of **HEPHASTUS** illustrated in Figure 1,Input sequences are first normalized via *RevIN* (Kim et al., 2022). The normalized data are processed by the *AMS-MoE* module, which decomposes the input into multiple scales using *Moving-Patch* and routes to top-$K$ experts via a learnable router. Each expert contains stacked *ST-Blocks*, where *Periodic Temporal Attention (PTA)* captures daily/weekly patterns using periodic embeddings, and *Heterogeneous Spatial Attention (HSA)* models spatial dependencies through node-specific transformations from a low-rank pattern library. The outputs of selected experts are aggregated and passed to an MLP predictor, followed by inverse RevIN for denormalization. The entire framework is end-to-end trainable, enabling adaptive, efficient, and structured spatio-temporal learning.

### 4.2 Adaptive Multi-Scale Mixture of Experts Architecture

To enable adaptive multi-scale modeling tailored to the intrinsic characteristics of each input time series, we design a **Adaptive Multi-Scale Mixture of Experts (AMS-MoE)** architecture. This framework consists of two main components: (1) **Temporal-aware routing mechanism** that dynamically assigns importance weights to multiple scale-specific "experts", and (2) **Adaptive aggregation module** that combines expert outputs according to these weights.

**Moving-Patch.** We propose an enhanced patching mechanism that improves upon existing approaches in (Nie et al., 2023) and (Chen et al., 2024) by introducing dynamic temporal context through overlapping patches with boundary preservation. Given an input sequence of length $H$, the method processes the data in three key phases:

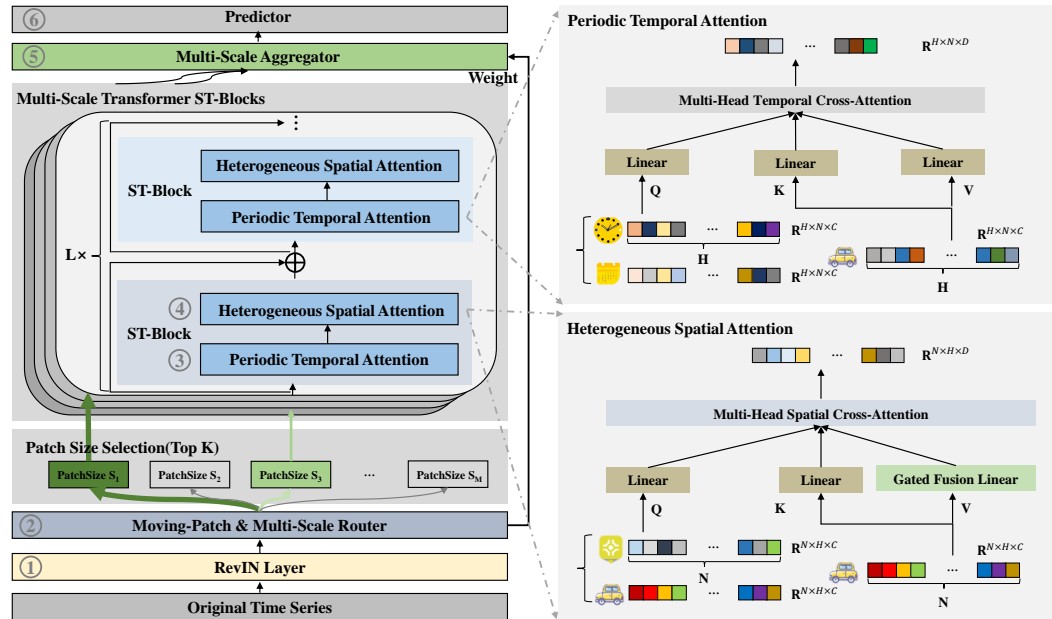

Figure 1: Overview of **HEPHASTUS**.

1. **Left-Padding with Boundary Replication**: To maintain temporal continuity at sequence boundaries, we extend the input by prepending $P-1$ copies of the initial element:

$$\tilde{\mathbf{X}} = [\underbrace{X_{t-H+1}, \ldots, X_{t-H+1}}_{P-1}, X_{t-H+1}, X_{t-H+2}, \ldots, X_t] \in \mathbb{R}^{N \times (H+P-1) \times d} \quad (2)$$

where $P$ is the patch size.

2. **Sliding Patch Extraction**: We then extract overlapping patches with stride 1 across the extended sequence:

$$\mathbf{p}_i = [\tilde{X}_i, \tilde{X}_{i+1}, \ldots, \tilde{X}_{i+P-1}] \in \mathbb{R}^{N \times P \times d} \quad \text{for} \quad i = 0, \ldots, H-1, \quad (3)$$

This generates $H$ patches $\mathcal{P} = \{\mathbf{p}_i\}_{i=0}^{H-1} \in \mathbb{R}^{N \times H \times P \times d}$ with dense temporal coverage.

3. **Linear Projection**: Apply learnable weight $W \in \mathbb{R}^{(P \cdot d) \times C}$ to flatten and project patches:

$$\mathcal{H} = \text{reshape}(\mathcal{P}) \cdot W \in \mathbb{R}^{N \times H \times C}, \quad (4)$$

where reshape$(\cdot)$ flattens last two dimensions.

**Temporal-Aware Expert Routing.** Each "expert" in our AMS-MoE corresponds to a Transformer encoder operating at a fixed patch size, representing one temporal resolution. To determine which scales are most relevant for the current input, we introduce a **temporal decomposition-based router**, which extracts key temporal features from the raw input sequence $\mathbf{X_t} \in \mathbb{R}^{N \times H \times C}$.

For each candidate patch size $p_i \in \{p_1, \ldots, p_M\}$, we apply the moving-patch operation and to combine these multi-scale representations through a gating mechanism:

$$\mathbf{X}_{\text{patch}} = \text{Softmax}(\mathcal{F}(\mathbf{X_t})) \cdot [\text{MovingPatch}_{p_1}(\mathbf{X_t}), \ldots, \text{MovingPatch}_{p_M}(\mathbf{X_t})], \quad (5)$$

where $\mathcal{F}(\cdot) : \mathbb{R}^{N \times H \times C} \to \mathbb{R}^{N \times M}$ is a learnable projection layer that produces fusion weights for each patch size. These components are concatenated with the original input and passed through a linear transformation to yield a compact temporal representation $\mathbf{X}_h \in \mathbb{R}^d$.

Based on $\mathbf{X}_h$, the router computes soft assignment probabilities over all available patch sizes (i.e., experts):

$$\mathcal{R}(\mathbf{X}_h) = \text{Softmax}\left(\mathbf{X}_h W_r + \epsilon \cdot \text{Softplus}(\mathbf{X}_h W_{\text{noise}})\right), \quad \epsilon \sim \mathcal{N}(0, 1), \quad (6)$$

where $W_r, W_{\text{noise}} \in \mathbb{R}^{d \times M}$ are trainable parameters, and $M$ denotes the total number of patch sizes. The injected noise encourages exploration during training and prevents premature convergence to suboptimal scales.

Finally, we enforce sparsity via Top-$K$ selection:

$$\mathcal{R}(\mathbf{X}_{\text{h}})_i = \begin{cases} R(\mathbf{X}_{\text{h}})_i & \text{if } i \in \text{TopK}(\mathcal{R}(\mathbf{X}_{\text{h}})) \\ 0 & \text{otherwise.} \end{cases} \tag{7}$$

This yields a sparse gating vector $\mathcal{R}(\mathbf{X}_{\text{h}}) \in \mathbb{R}^M$, assigning non-zero weights only to the top-$K$ selected scales.

**Multi-Scale Output Aggregation.** Let $\mathbf{X}_{\text{t}}^i = \text{MovingPatch}_{p_i}(\mathbf{X}_{\text{t}}) \in \mathbb{R}^{N \times H \times C}$ denote the $i$-th Moving-Patch branch with patch size $S_i$. $E_i$ denote the $i$-th ST-Block. The final aggregated output is computed as:

$$\mathbf{X}_{\text{out}} = \sum_{i=1}^{M} \mathcal{S}\left(\bar{R}(\mathbf{X}_{\text{h}})_i > 0\right) \cdot \bar{R}(\mathbf{X}_{\text{h}})_i \cdot E_i(\mathbf{X}_{\text{t}}^i), \tag{8}$$

where $\mathcal{S}(\cdot)$ is the indicator function ensuring only activated experts contribute to the output.

This formulation mirrors classical MoE designs but extends it to the multi-scale time-series domain by incorporating temporal-aware routing and dynamic alignment of heterogeneous outputs.

### 4.3 PERIODIC TEMPORAL ATTENTION MECHANISM

Traffic flow data exhibits strong periodic patterns, especially on a daily and weekly basis. To better capture such temporal regularities in the self-attention mechanism, we introduce an enhanced time-aware attention module that incorporates learnable periodic embeddings.

Let $\mathcal{H} \in \mathbb{R}^{N \times H \times C}$ be the input. We define two learnable periodic embedding matrices:

$$P_D \in \mathbb{R}^{L_D \times d_{\text{tid}}}, \quad P_W \in \mathbb{R}^{L_W \times d_{\text{tiw}}}, \tag{9}$$

where $L_D = 288$ (5-minute intervals per day), $L_W = 288 \times 7$, and $d_{\text{tid}}, d_{\text{tiw}}$ are embedding dimensions. For each time step $t$, we retrieve embeddings based on $t \mod L_D$ and $t \mod L_W$, forming a concatenated periodic embedding:

$$P_e = [P_D[t \mod L_D, :] \;\|\; P_W[t \mod L_W, :]]_{t=1}^{H} \in \mathbb{R}^{H \times (d_{\text{tid}} + d_{\text{tiw}})}. \tag{10}$$

Next, we apply a linear projection using a trainable weight matrix $W_{qt} \in \mathbb{R}^{(d_{tid} + d_{tiw}) \times D}$ to transform $P_e$ into a query-compatible representation:$P_e' = P_e W_{qt} \in \mathbb{R}^{H \times D}$.This projected embedding is then broadcast across all spatial nodes to form the final time-aware query tensor:$Q_t = \text{Broadcast}(P_e') \in \mathbb{R}^{N \times H \times D}$.The key $K_t$ and value $V_t$ tensors are computed following standard self-attention procedures via learnable projections:

$$K_t = \mathcal{H} W_{kt}, \quad V_t = \mathcal{H} W_{vt}, \tag{11}$$

where $W_{kt}, W_{vt} \in \mathbb{R}^{C \times D}$ are learnable projection matrices, resulting in $K_t, V_t \in \mathbb{R}^{N \times H \times D}$.The cross-attention score are subsequently computed as:

$$A_t = \text{softmax}\left(\frac{Q_t K_t^\top}{\sqrt{D}}\right), \tag{12}$$

with $A_t \in \mathbb{R}^{N \times H \times H}$ encoding temporal dependencies across distinct spatial nodes. The output of the periodic temporal transformer, denoted $Z_t \in \mathbb{R}^{N \times H \times D}$, is then obtained via $Z_t = A_t V_t$.We further incorporate layer normalization, residual connections, and a multi-head mechanism in the design.

## 4.4 HETEROGENEOUS SPATIAL ATTENTION MECHANISM

To address the spatial indistinguishability problem in traffic forecasting, we propose a novel spatial attention mechanism that explicitly models both shared and node-specific patterns across the road network. Let $\mathcal{H} \in \mathbb{R}^{N \times H \times C}$ denote the input spatiotemporal feature tensor. First, we introduce a learnable node embedding matrix $S_e \in \mathbb{R}^{N \times r}$, which is randomly initialized and updated end-to-end during training. This matrix captures intrinsic characteristics of each node. To incorporate temporal dynamics, we broadcast $S_e$ across the time dimension to obtain $\tilde{S}_e \in \mathbb{R}^{N \times H \times r}$.

The query representation $Q_s$ is computed by first applying a linear transformation on $\mathcal{H}$:

$$Q' = \mathcal{H}W_{qs1}, \quad W_{qs1} \in \mathbb{R}^{C \times d_q}, \tag{13}$$

where $Q' \in \mathbb{R}^{N \times H \times d_q}$. Then, we concatenate $Q'$ with $\tilde{S}_e$, followed by another linear projection:

$$Q_s = [\tilde{S}_e \parallel Q']W_{qs2}, \quad W_{qs2} \in \mathbb{R}^{(r+d_q) \times D}, \tag{14}$$

resulting in $Q_s \in \mathbb{R}^{N \times H \times D}$.

Next, we design two complementary value representations: one for capturing shared global patterns ($V_{sc}$) and another for modeling node-specific behaviors ($V_{ss}$).

**Common Linear.** The shared component is obtained via a globally shared weight matrix:

$$V_{sc} = \mathcal{H}W_{vc} \in \mathbb{R}^{N \times H \times D}, \quad W_{vc} \in \mathbb{R}^{C \times D}. \tag{15}$$

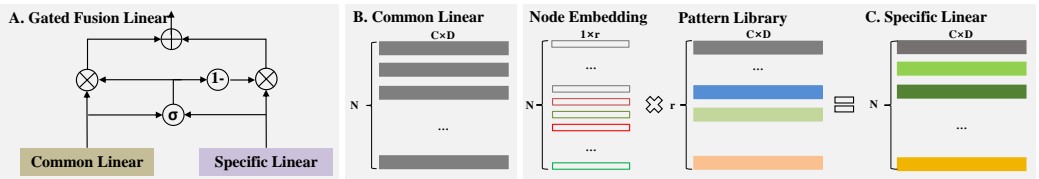

Figure 2: The structure of the **Gated Fusion Linear**.

**Specific Linear.** For the node-specific component, directly assigning an individual transformation matrix per node would result in excessive parameterization ($O(N \times C \times D)$). Instead, inspired by tensor decomposition techniques, we decompose the full parameter tensor into low-rank components. Specifically, we define a pattern library tensor $PL \in \mathbb{R}^{r \times C \times D}$ and reuse the previously defined node embeddings $S_e \in \mathbb{R}^{N \times r}$ to generate adaptive transformations through tensor multiplication:

$$W_{vs}^{(i)} = \sum_{k=1}^{r} S_e[i, k] \cdot PL[k, :, :] \in \mathbb{R}^{C \times D}, \quad \forall i \in [N]. \tag{16}$$

This yields a set of node-adaptive matrices $\{W_{vs}^{(i)}\}_{i=1}^{N}$, enabling efficient yet expressive node-specific mappings:

$$V_{ss}^{(i)} = \mathcal{H}[i, :, :]W_{vs}^{(i)}, \quad \forall i \in [N], \tag{17}$$

which are concatenated to form $V_{ss} \in \mathbb{R}^{N \times H \times D}$.

**Gated Fusion.** We then dynamically fuse $V_{sc}$ and $V_{ss}$ using a gating mechanism. Concatenating these along the channel axis and passing through a sigmoid function produces $\lambda \in [0, 1]$:

$$\lambda = \sigma([V_{sc} \parallel V_{ss}]W_g), \quad W_g \in \mathbb{R}^{2D \times 1}, \tag{18}$$

To enable element-wise modulation, we expand $\lambda$ into a full-sized gate tensor $\Lambda \in \mathbb{R}^{N \times H \times D}$ by broadcasting it across the spatial and temporal dimensions. The final fused value representation is then computed via the Hadamard product:$V_s = \Lambda \circ V_{sc} + (1 - \Lambda) \circ V_{ss}$.where $\circ$ denotes the Hadamard (element-wise) product, and $1$ is a tensor of ones with the same shape as $\Lambda$.

Finally, the key representation $K_s$ is derived similarly to standard self-attention via $K_s = \mathcal{H}W_{ks}$, With all components ready, we compute spatial cross-attention along the spatial axis:

$$A_s = \text{softmax}\left(\frac{Q_s K_s^\top}{\sqrt{D}}\right), \tag{19}$$

with $A_s \in \mathbb{R}^{H \times N \times N}$ encoding spatial dependencies across distinct time frames. The output of the heterogeneous spatial transformer, denoted $Z_s \in \mathbb{R}^{N \times H \times D}$, is then obtained via $Z_s = A_s V_s$. We further incorporate layer normalization, residual connections, and a multi-head mechanism in the design.

### 4.5  Loss Function

We adopt MAE as the prediction loss function:

$$\mathcal{L}_{\text{p}} = MAE\left(X_{t+1:t+F}, \hat{X}_{t+1:t+F}\right) = \frac{1}{NF}\sum_{i=1}^{F}\sum_{j=1}^{N}\left|X_{i,j} - \hat{X}_{i,j}\right|. \tag{20}$$

When training models with a Mixture-of-Experts (MoE) architecture, optimizing solely for prediction error often induces severe load imbalance across experts. A well-documented issue is *routing collapse* (Shazeer et al., 2017), wherein the router converges to consistently selecting only a small subset of experts, thereby starving the remaining experts of training signals. To counteract this, we adopt an auxiliary balancing loss following the methodology of (Fedus et al., 2022; Dai et al., 2024), which explicitly encourages uniform utilization across all experts and mitigates routing collapse.:

$$\mathcal{L}_{\text{aux}} = M\sum_{i=1}^{M}f_i r_i, \quad f_i = \frac{1}{KT}\sum_{t=1}^{T}\mathbb{I}(\text{Expert } i \text{ selected at } t), \quad r_i = \frac{1}{T}\sum_{t=1}^{T}s_{i,t}. \tag{21}$$

Here, $f_i$ denotes the token assignment frequency to expert $i$, while $r_i$ represents the mean router probability for that expert.

Combining the prediction loss and auxiliary loss functions yields the model's total loss function as:

$$\mathcal{L} = \mathcal{L}_{\text{p}} + \alpha\mathcal{L}_{\text{aux}}. \tag{22}$$

## 5  Experiments

### 5.1  Experimental Setup

***Datasets.*** We evaluate our method on six real-world traffic forecasting benchmarks: METR-LA, PEMS-BAY, PEMS03, PEMS04, PEMS07, and PEMS08, covering both traffic speed and volume prediction tasks. These datasets provide sensor graphs to model spatial dependencies. For reproducibility and fair comparison, all experiments are conducted on the BasicTS (Liang et al., 2022) platform. Further dataset statistics and preprocessing details are provided in Appendix A.2.1.

***Implementation.*** Our model is implemented in PyTorch and trained on a single GeForce RTX 4090 GPU. We use standard train/validation/test splits. The model is optimized using Adam with a decaying learning rate and early stopping. Detailed hyperparameter settings, training configurations, and optimization strategies are included in Appendix A.2.2.

***Metrics.*** We employ three standard evaluation metrics for traffic forecasting, i.e, MAE, RMSE and MAPE. Following previous work, we select the average performance of all predicted 12 horizons on the PEMS03, PEMS04, PEMS07 and PEMS08 datasets. To evaluate the METR-LA and PEMS-BAY datasets, we compare the performance on horizon 3, 6 and 12 (15, 30, and 60 min).

***Baselines.*** We compare against a comprehensive set of baselines, including STGNNs, Identity-aware models, and Transformer-based models. Selected models include STGCN (Yu et al., 2018),

Table 1: Performance on METR-LA and PEMS-BAY.

| Datasets | Metrics | STGCN | DCRNN | GWNet | ASTGCN | AGCRN | MTGNN | DGCRN | STID | STAEformer | GMAN | PDFormer | STWave | iTransformer | PatchTST | PathFormer | HEPHASTUS |
|---|---|---|---|---|---|---|---|---|---|---|---|---|---|---|---|---|---|
| METR-LA Horizon 3 (15 min) | MAE | 2.73 | 2.65 | 2.65 | 3.08 | 2.86 | 2.68 | 2.65 | 2.83 | 2.71 | 2.79 | 2.83 | 2.82 | 3.21 | 3.38 | 3.12 | **2.62** |
| | RMSE | 5.28 | 5.19 | 5.21 | 6.12 | 5.51 | 5.15 | 5.18 | 5.56 | 5.23 | 5.52 | 5.45 | 5.57 | 7.16 | 7.33 | 6.93 | **5.13** |
| | MAPE | 7.07% | 6.93% | 6.99% | 8.16% | 7.26% | 6.89% | 6.86% | 7.78% | 6.96% | 7.46% | 7.77% | 7.51% | 8.38% | 8.69% | 7.96% | **6.76%** |
| Horizon 6 (30 min) | MAE | 3.21 | 3.06 | 3.12 | 3.79 | 3.22 | 3.08 | 3.06 | 3.19 | 3.06 | 3.16 | 3.2 | 3.21 | 4.15 | 4.29 | 3.96 | **2.93** |
| | RMSE | 6.36 | 6.31 | 6.26 | 7.62 | 6.48 | 6.16 | 6.19 | 6.57 | 6.18 | 6.53 | 6.46 | 6.66 | 8.66 | 9.46 | 7.85 | **6.05** |
| | MAPE | 8.68% | 8.42% | 8.39% | 10.65% | 8.99% | 8.21% | **8.11%** | 9.39% | 8.41% | 8.69% | 9.19% | 9.31% | 9.98% | 11.33% | 10.55% | 8.12% |
| Horizon 12 (60 min) | MAE | 3.56 | 3.55 | 3.55 | 4.95 | 3.62 | 3.49 | 3.42 | 3.55 | 3.43 | 3.46 | 3.62 | 3.56 | 5.55 | 5.82 | 5.06 | **3.34** |
| | RMSE | 7.39 | 7.36 | 7.26 | 9.81 | 7.46 | 7.26 | 7.18 | 7.55 | 7.21 | 7.23 | 7.47 | 7.54 | 11.29 | 12.39 | 9.92 | **7.06** |
| | MAPE | 10.32% | 10.34% | 9.98% | 14.92% | 10.27% | 9.81% | 9.79% | 10.95% | 10.09% | 10.05% | 10.91% | 10.81% | 15.51% | 15.74% | 15.09% | **9.76%** |
| PEMS-BAY Horizon 3 (15 min) | MAE | 1.36 | 1.32 | 1.31 | 1.45 | 1.35 | 1.36 | 1.33 | 1.31 | 1.31 | 1.35 | 1.32 | 1.33 | 1.47 | 1.49 | 1.43 | **1.31** |
| | RMSE | 2.86 | 2.81 | 2.83 | 3.15 | 2.88 | 2.81 | **2.79** | 2.81 | 2.80 | 2.88 | 2.83 | 2.85 | 3.20 | 3.28 | 3.12 | 2.83 |
| | MAPE | 2.89% | 2.73% | **2.68%** | 3.05% | 2.93% | 2.83% | 2.82% | 2.78% | 2.77% | 2.88% | 2.78% | 2.81% | 3.08% | 3.07% | 3.06% | 2.71% |
| Horizon 6 (30 min) | MAE | 1.72 | 1.68 | **1.63** | 1.98 | 1.68 | 1.66 | 1.67 | 1.64 | 1.66 | 1.67 | 1.64 | 1.65 | 2.03 | 2.06 | 2.01 | 1.65 |
| | RMSE | 3.84 | 3.75 | 3.82 | 4.63 | 3.77 | 3.79 | 3.78 | 3.73 | 3.73 | 3.73 | 3.71 | 3.86 | 4.81 | 4.85 | 4.82 | **3.68** |
| | MAPE | 3.86% | 3.73% | 3.71% | 4.46% | 3.75% | 3.79% | 3.72% | 3.73% | 3.71% | 3.81% | 3.71% | 3.69% | 4.41% | 4.47% | 4.38% | **3.66%** |
| Horizon 12 (60 min) | MAE | 2.12 | 1.99 | 2.01 | 2.72 | 1.96 | 1.93 | 1.92 | 1.91 | 1.89 | 1.97 | 1.91 | 1.92 | 2.85 | 2.92 | 2.76 | **1.88** |
| | RMSE | 4.65 | 4.53 | 4.61 | 6.25 | 4.53 | 4.46 | 4.43 | 4.42 | 4.41 | 4.49 | 4.43 | 4.39 | 6.79 | 6.87 | 6.63 | **4.38** |
| | MAPE | 4.81% | 4.89% | 4.67% | 6.25% | 4.56% | 4.61% | 4.52% | 4.55% | 4.53% | 4.54% | 4.51% | 4.51% | 6.52% | 6.67% | 6.26% | **4.46%** |

DCRNN (Li et al., 2018), GWNet (Wu et al., 2019), ASTGCN (Guo et al., 2019), AGCRN (Bai et al., 2020), MTGNN (Wu et al., 2020), DGCRN (Li et al., 2023), STID (Shao et al., 2022a), STAEformer (Liu et al., 2023a), GMAN (Zheng et al., 2020), PDFormer (Jiang et al., 2023), STWave (Fang et al., 2023), iTransformer (Liu et al., 2023c), PatchTST (Nie et al., 2023), and PathFormer (Chen et al., 2024). A complete description of baseline selection and implementation is provided in Appendix A.2.3.

## 5.2 Overall Performance

The results summarized in Table 1 and Table 2. **HEPHASTUS** achieves state-of-the-art performance across six real-world traffic forecasting benchmarks, consistently outperforming a wide range of baseline methods, including classical spatio-temporal graph neural networks (STGNNs), attention-based models, and recent patch-based or identity-aware architectures. Key findings include: (1) Consistent performance gains across both speed and volume datasets, and scalability to networks of varying sizes, from 170 to 883 sensors; (2) Significant advantages over patch-based models like PatchTST and PathFormer, which suffer from lack of spatial awareness and rigid patching; HEPHASTUS uses adaptive multi-scale routing and heterogeneous fusion for dynamic, input-aware scale selection and integrated spatio-temporal modeling. (3) Outperformance of strong STGNNs (e.g., GWNet, AGCRN, MTGNN) and dynamic models (e.g., DGCRN), highlighting the effectiveness of jointly modeling adaptive temporal granularity, explicit periodicity, and dynamic spatial heterogeneity, achieving higher accuracy with lower parametric complexity through structured priors. (4) STID and STAEformer decouple the dependency on predefined graphs by encoding node locations and temporal patterns into vectors, achieving certain competitiveness in some scenarios. However, their ability to model dynamic spatio-temporal patterns remains insufficient, as evidenced by their relatively poor performance on the METR-LA and PEMS03 datasets. Overall, HEPHASTUS demonstrates that a unified, adaptive, and structured approach is essential for capturing the complexity of urban traffic, with its core innovations proving both effective and necessary across diverse forecasting scenarios.For a more detailed discussion of the overall performance, please refer to the Appendix A.3.

Table 2: Performance on PEMS03, PEMS04, PEMS07, and PEMS08.

| Datasets | Metrics | STGCN | DCRNN | GWNet | ASTGCN | AGCRN | MTGNN | DGCRN | STID | STAEformer | GMAN | PDFormer | STWave | iTransformer | PatchTST | PathFormer | HEPHASTUS |
|---|---|---|---|---|---|---|---|---|---|---|---|---|---|---|---|---|---|
| PEMS03 | MAE | 15.81 | 15.49 | **14.52** | 16.53 | 15.31 | 14.89 | 14.63 | 15.33 | 15.38 | 14.83 | 15.16 | 15.06 | 19.26 | 19.89 | 19.01 | 14.76 |
| | RMSE | 27.53 | 27.09 | 25.28 | 29.16 | 26.59 | 25.36 | 25.41 | 27.59 | 25.45 | 26.32 | 26.57 | 25.57 | 31.06 | 32.63 | 30.15 | **25.21** |
| | MAPE | 16.15% | 15.56% | 15.46% | 16.65% | 15.85% | 15.16% | 14.99% | 16.43% | 15.97% | 15.68% | 15.86% | 15.56% | 19.31% | 19.63% | 19.23% | **14.58%** |
| PEMS04 | MAE | 19.73 | 19.65 | 18.82 | 20.15 | 19.32 | 19.23 | 19.12 | 18.35 | 18.33 | 19.28 | 18.96 | 18.42 | 23.86 | 27.25 | 21.95 | **18.21** |
| | RMSE | 31.56 | 31.15 | 30.06 | 31.78 | 30.96 | 31.08 | 30.52 | 29.86 | 30.23 | 31.12 | 30.64 | 30.16 | 37.26 | 42.86 | 34.26 | **29.83** |
| | MAPE | 13.52% | 13.41% | 13.14% | 14.12% | 13.17% | 13.26% | 13.05% | 12.53% | 12.39% | 13.19% | 12.88% | 12.53% | 16.26% | 17.94% | 15.99% | **12.01%** |
| PEMS07 | MAE | 22.19 | 21.19 | 20.52 | 23.98 | 20.46 | 21.26 | 20.18 | 19.91 | 19.90 | 21.25 | 19.91 | 19.90 | 27.32 | 29.81 | 27.26 | **19.18** |
| | RMSE | 35.86 | 34.23 | 33.43 | 38.85 | 34.56 | 34.78 | 33.64 | 32.69 | 32.73 | 34.21 | 33.53 | 33.55 | 45.69 | 47.11 | 43.26 | **32.65** |
| | MAPE | 9.46% | 9.11% | 8.86% | 10.26% | 8.89% | 8.95% | 8.78% | 8.31% | 8.26% | 9.02% | 8.52% | 8.39% | 12.06% | 12.70% | 11.83% | **8.13 %** |
| PEMS08 | MAE | 16.26 | 15.26 | 14.69 | 17.52 | 15.68 | 15.26 | 14.93 | 14.21 | 14.02 | 15.07 | 14.81 | 13.89 | 18.68 | 20.79 | 18.25 | **13.56** |
| | RMSE | 25.48 | 24.37 | 23.62 | 27.16 | 24.81 | 24.52 | 23.96 | 23.35 | 23.41 | 24.23 | 23.75 | 23.81 | 32.19 | 33.75 | 31.49 | **23.39** |
| | MAPE | 10.86% | 10.19% | 9.58% | 11.29% | 10.46% | 10.63% | 9.81% | 9.32% | 9.28% | 9.96% | 9.56% | 9.21% | 12.13% | 12.78% | 11.98% | **8.96%** |

## 5.3 Ablation Study

To validate the contribution of each component in **HEPHASTUS**, we conduct a comprehensive ablation study on METR-LA, PEMS-BAY, PEMS04, and PEMS08, with results summarized in

Table 3.We consider the following variants: **w/o AMS-MoE** (single fixed patch size $P = 3$), **w/o PTA** (standard temporal attention without periodic encodings), **w/o HSA** (shared value projection, no node-specific spatial modeling), **Fixed-Scale MoE** (multi-scale experts with uniform routing weights), and **Noisy-Routing Only** (router without Top-$K$ sparsification).

Results show that the full model achieves the best performance across all metrics and datasets. Removing AMS-MoE leads to the largest performance drop, confirming the importance of input-adaptive multi-scale modeling. The degradation from removing PTA or HSA highlights the value of explicit periodicity modeling and dynamic spatial heterogeneity learning. Notably, *Fixed-Scale MoE* underperforms the full model, demonstrating that adaptive scale selection is critical for capturing diverse temporal patterns. *Noisy-Routing Only* performs better than non-adaptive variants but worse than full routing, suggesting that sparsity enhances efficiency without sacrificing expressiveness. These results confirm that each component is necessary and that their combination yields synergistic gains. See Appendix A.4 for full details.

Table 3: Ablation study on METR-LA, PEMS-BAY, PEMS04, and PEMS08.

| Model | METR-LA | | | PEMS-BAY | | | PEMS04 | | | PEMS08 | | |
|---|---|---|---|---|---|---|---|---|---|---|---|---|
| | MAE | RMSE | MAPE(%) | MAE | RMSE | MAPE(%) | MAE | RMSE | MAPE(%) | MAE | RMSE | MAPE(%) |
| HEPHASTUS (Full) | **3.36** | **7.06** | **9.76** | **1.88** | **4.38** | **4.46** | **18.21** | **29.83** | **12.01** | **13.56** | **23.39** | **8.96** |
| w/o AMS-MoE | 3.51 | 7.28 | 10.21 | 1.96 | 4.57 | 4.72 | 19.12 | 30.78 | 12.65 | 14.49 | 24.37 | 9.78 |
| w/o PTA | 3.45 | 7.19 | 10.03 | 1.92 | 4.48 | 4.61 | 18.78 | 30.31 | 12.38 | 13.98 | 23.85 | 9.42 |
| w/o HSA | 3.48 | 7.21 | 10.15 | 1.94 | 4.51 | 4.58 | 18.96 | 30.45 | 12.41 | 14.12 | 24.03 | 9.51 |
| Fixed-Scale MoE | 3.47 | 7.23 | 10.08 | 1.93 | 4.49 | 4.55 | 18.83 | 30.26 | 12.27 | 14.01 | 23.92 | 9.34 |
| Noisy-Routing Only | 3.41 | 7.15 | 9.92 | 1.90 | 4.43 | 4.50 | 18.54 | 30.09 | 12.18 | 13.78 | 23.61 | 9.24 |

## 5.4 PARAMETER SENSITIVITY ANALYSIS

We evaluate the sensitivity of **HEPHASTUS** to three key hyperparameters on METR-LA and PEMS08, with results summarized in Figure 3. The number of experts $M$ governs the model's capacity to capture multi-scale temporal patterns. When $M$ is too small (e.g., 2), the model lacks sufficient scale diversity to differentiate transient fluctuations from longer-term dynamics. Increasing $M$ to 4 enables the adaptive router to assign specialized experts more precisely according to input characteristics, enhancing representational flexibility. Further increasing $M$ leads to expert redundancy and routing instability, with diminishing or even slightly negative returns, indicating that $M = 4$ strikes an optimal balance between expressiveness and efficiency.

The top-$K$ sparsity factor $K$ controls the activation density in dynamic routing, directly affecting generalization. With $K = 1$, the routing is overly sparse, limiting collaboration among experts and creating information bottlenecks. $K = 2$ achieves an effective trade-off—retaining the regularization benefits of sparsity while allowing sufficient expert interaction for robust feature integration. Larger $K$ (e.g., 4) approaches dense activation, weakening the selectivity of routing and increasing susceptibility to overfitting, particularly in noisy or complex scenarios. Meanwhile, the rank dimension $r$ of the pattern library in HSA determines the capacity to model heterogeneous spatial dependencies. A small $r$ constrains the diversity of learnable spatial patterns, limiting discrimination among functionally distinct nodes. At moderate $r$ (e.g., 8), the low-rank structure captures dominant spatial heterogeneities effectively, with computational and parameter costs significantly lower than node-wise modeling. Performance plateaus beyond this point, confirming that HSA achieves strong efficiency and scalability. Overall, **HEPHASTUS** demonstrates robustness across settings, validating its design of adaptive, sparse, and low-rank mechanisms for scalable spatio-temporal forecasting. More detailed analysis can be found in the Appendix A.5.

## 5.5 EFFICIENCY STUDY

To comprehensively evaluate the practical deployment potential of **HEPHASTUS**, we conduct a systematic efficiency comparison with state-of-the-art baselines on the PEMS04 dataset(batch size $B$=16). As shown in Table 4, we analyze multiple efficiency metrics including parameter count, training/inference time, and GPU memory usage.

As shown in the table, models using TCNs, such as STGCN, GWNet, perform relatively well because of the high computational efficiency of convolution. In comparison, the next three models

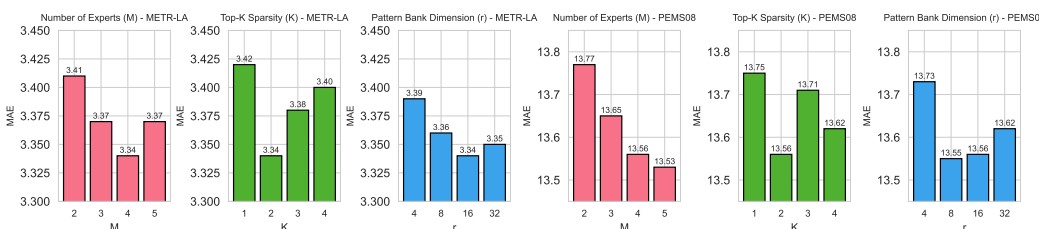

Figure 3: Parameter Sensitivity Analysis on METR-LA and PEMS08.

with GCRU architecture show a notable efficiency gap due to the iterative nature of RNNs that intrinsically presents a disadvantage. Additionally, Transformer-based models like STAEformer, STWave, PDFormer and PathFormer are even worse due to their quadratic computational complexity. Memory consumption analysis reveals that **HEPHASTUS** utilizes 5475MB GPU memory, which is lower than PDFormer (8295MB) and PathFormer (6652MB). This reduction in memory footprint, while maintaining competitive predictive performance, highlights our architectural optimizations for resource-constrained environments.

The efficiency analysis demonstrates that HEPHASTUS strikes an effective balance between model capacity and computational requirements, making it suitable for both research and practical deployment scenarios where computational resources are a consideration.

Table 4: Efficiency comparison on PEMS04 dataset.

| Model | #Params | Train Time / Iteration | Train Time / Epoch | Inference Time / Iteration | Inference Time / Epoch | GPU Memory Usage |
|---|---|---|---|---|---|---|
| STGCN (Yu et al., 2018) | 297K | 21ms | 13.12s | 11ms | 2.28s | 1090MB |
| GWNet (Wu et al., 2019) | 311K | 37ms | 23.49s | 13ms | 2.79s | 1078MB |
| DCRNN (Li et al., 2018) | 372K | 153ms | 97.3s | 68ms | 14.43s | 2216MB |
| AGCRN (Bai et al., 2020) | 749K | 72ms | 45.51s | 27ms | 5.63s | 2226MB |
| DGCRN (Li et al., 2023) | 207K | 235ms | 149.18s | 183ms | 38.67s | 1898MB |
| STAEformer Liu et al. (2023a) | 1355K | 71ms | 45.04s | 26ms | 5.44s | 4666MB |
| STWave Fang et al. (2023) | 882K | 68ms | 43.25s | 30ms | 6.37s | 3190MB |
| PDFormer (Jiang et al., 2023) | 1263K | 81ms | 51.72s | 34ms | 7.18s | 8295MB |
| PathFormer (Chen et al., 2024) | 1133K | 156ms | 99.47s | 39ms | 8.13s | 6652MB |
| **HEPHASTUS** | 716K | 98ms | 62.07s | 31ms | 6.49s | 5475MB |

## 5.6 CASE STUDY

To qualitatively validate the adaptive routing behavior of AMS-MoE, we visualize two sample predictions from PEMS-BAY and PEMS04. As illustrated in Figure 4, during peak hours with rapid traffic variations (e.g., morning and evening rush), the router predominantly selects experts with smaller patch sizes, enabling fine-grained capture of transient fluctuations. In contrast, during off-peak periods with smoother traffic patterns, the router shifts preference to experts with larger patch sizes, emphasizing long-term trend modeling. This input-dependent scale selection demonstrates AMS-MoE's ability to dynamically align temporal granularity with traffic dynamics, enhancing both local detail preservation and global pattern extraction.More detailed analysis can be found in the Appendix A.6.

## 6 CONCLUSION

In this study, we focus on multi-scale modeling technique for traffic time series forecasting, i.e., multi-scale temporal modeling, MoE-based dynamic routing. Our model achieve the SOTA performance on six traffic benchmarks. Further studies demonstrate that our model can effectively capture intrinsic spatio-temporal dependencies. Instead of designing complicated models, our study shows a promising direction for addressing the challenges in traffic forecasting.

## ETHICS STATEMENT

This research presents a traffic forecasting methodology with potential applications in intelligent transportation systems. While our work focuses on improving predictive accuracy, we recognize several ethical considerations:

**Data Privacy:** All datasets used in this study are publicly available and contain only aggregated, anonymized traffic measurements without personally identifiable information.

**Societal Impact:** Accurate traffic forecasting can contribute to reduced congestion, lower emissions, and improved urban mobility. However, deployment in real-world systems requires careful consideration of fairness, as biased predictions could disproportionately affect certain communities. We encourage practitioners to validate model performance across diverse geographic and socioeconomic contexts.

**Misuse Potential:** While traffic prediction itself is benign, the technology could potentially be misused for surveillance or traffic control that restricts mobility. We advocate for transparent deployment with appropriate oversight mechanisms.

## REPRODUCIBILITY STATEMENT

To ensure reproducibility, we provide the following:

**Code Availability:** Our implementation will be made publicly available upon acceptance.

**Datasets:** All datasets (METR-LA, PEMS-BAY, PEMS03/04/07/08) are publicly accessible through established benchmarks.

**Experimental Details:**

- Complete hyperparameter settings and training configurations are provided in the appendix
- We use standard train/validation/test splits consistent with prior work
- All experiments are conducted on the BasicTS platform for fair comparison
- Evaluation metrics (MAE, RMSE, MAPE) follow community standards

**Computational Resources:** Experiments were run on a single GeForce RTX 4090 GPU, with training times typically under 12 hours per dataset.

We are committed to supporting reproducibility and will release code, configuration files, and detailed instructions to facilitate replication of our results.

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

# A APPENDIX

## A.1 USE OF LARGE LANGUAGE MODELS

In the preparation of this manuscript, the authors employed a Large Language Model (LLM) solely for the purpose of language polishing and refinement. The LLM was utilized exclusively to improve the clarity, coherence, and grammatical accuracy of the text, ensuring that the academic content remains entirely original and authored by the human contributors.

No generative use of the LLM was made in the formation of scientific concepts, methodological design, data analysis, interpretation of results, or drafting of technical content. All ideas, hypotheses, experiments, and conclusions are the product of the authors' intellectual contribution.

The use of the LLM was supervised and reviewed by the authors to guarantee the integrity and authenticity of the scholarly work presented in this paper.

## A.2 EXPERIMENTAL DETAILS

### A.2.1 DATASETS

***Datasets.*** Our method is verified on six traffic forecasting benchmarks, i.e., METR-LA, PEMS-BAY, PEMS03, PEMS04, PEMS07, and PEMS08. The first two datasets are traffic speed datasets recorded every 5 minutes. The datasets include sensor graphs to indicate spatial dependencies between sensors (Li et al., 2018). The last four datasets are traffic flow datasets recorded every 5 minutes. These datasets include sensor graphs to indicate dependencies between sensors (Guo et al., 2019). More details are shown in Table 5.

### A.2.2 IMPLEMENTATION

**Implementation Framework.** Our model is implemented using PyTorch and trained on a Linux server equipped with a GeForce RTX 4090 GPU. All experiments are conducted on the BasicTS platform (Liang et al., 2022) to ensure fair and consistent comparisons with existing methods.

**Data Splits.** We follow standard dataset partitioning protocols: METR-LA and PEMS-BAY use a 7:1:2 ratio for training, validation, and test sets, while PEMS03, PEMS04, PEMS07, and PEMS08 employ a 6:2:2 split.

**Hyperparameters.** The model demonstrates robustness to hyperparameter variations. Key settings include:

- Embedding dimensions: $d_{tid} = 16$ and $d_{tiw} = 32$ for temporal embeddings, $r = 16$ for spatial embeddings
- Stacked ST-Blocks: Each expert contains $L = 3$ ST-Block
- Number of attention heads: 4
- Input and prediction length: $H = F = 12$ (1-hour windows at 5-minute intervals)
- Optimizer: Adam with initial learning rate 0.001 and exponential decay
- Batch size: 64
- Early stopping: Triggered if validation loss plateaus for 30 consecutive epochs

**AMS-MoE Configuration.** The Adaptive Multi-Scale Mixture of Experts uses $M = 4$ experts with patch sizes $P \in \{1, 2, 3, 6\}$ corresponding to 5-minute, 10-minute, 15-minute, and 30-minute temporal scales. The router selects top-$K = 2$ experts per input sequence.

### A.2.3 BASELINES

***Baselines.*** Our study incorporates a comprehensive set of baselines with publicly available official implementations, encompassing traditional approaches, representative deep learning methods, and the most recent state-of-the-art works. For spatial-temporal graph neural networks (STGNNs),

we evaluate STGCN (Yu et al., 2018), DCRNN (Li et al., 2018), GWNet (Wu et al., 2019), AST-GCN (Guo et al., 2019), AGCRN (Bai et al., 2020), MTGNN (Wu et al., 2020) and DGCRN (Li et al., 2023). STID (Shao et al., 2022a) and STAEformer (Liu et al., 2023a) eliminate complex graph neural modules by injecting learnable spatial and temporal identity embeddings, treating each sensor's time series as independent while still achieving competitive performance. We include them as strong baselines because they represent a paradigm shift toward parameter-efficient, graph-free architectures, offering a direct contrast to our approach that explicitly models dynamic spatial dependencies. Although Transformer-based approaches like Informer (Zhou et al., 2021), Pyraformer (Liu et al., 2022), FEDformer (Zhou et al., 2022), and Autoformer (Wu et al., 2021) exist for time series forecasting, they are not specifically designed for short-term traffic prediction. Therefore, we select GMAN (Zheng et al., 2020) and PDFormer (Jiang et al., 2023) as Transformer models specifically developed for our target task. Since our model employs patching operations on temporal data, we also include PatchTST (Nie et al., 2023) and PathFormer (Chen et al., 2024) as comparative baselines.All baselines are evaluated using their official implementations and default or recommended hyperparameters from the BasicTS library to ensure fair comparisons.

## A.3 OVERALL PERFORMANCE

Our extensive evaluation across six real-world traffic forecasting benchmarks, spanning diverse urban scales, sensor densities, and traffic dynamics, demonstrates that **HEPHASTUS** consistently achieves state-of-the-art performance, outperforming a wide spectrum of baselines including classical STGNNs, attention-based architectures, and recent patch-based models. The results summarized in Table 1 and Table 2.

First, **the performance gain is consistent across both speed and flow datasets**, as well as under varying network sizes from 170 to 883 sensors. This indicates that HEPHASTUS generalizes well and is robust to spatial scale and data modality.

Second, **HEPHASTUS significantly outperforms recent patch-based methods** such as PatchTST and PathFormer—despite their success in general time series forecasting. On all datasets, these models exhibit notably higher errors, likely due to their lack of explicit spatial modeling and fixed patching strategies that fail to adapt to spatio-temporal dynamics. In contrast, our adaptive multi-scale routing and heterogeneous fusion enable input-aware scale selection and structured interaction between temporal and spatial dimensions.

Third, **the superiority over strong STGNNs (e.g., GWNet, AGCRN, MTGNN) and dynamic models (e.g., DGCRN)** underscores the importance of jointly modeling adaptive temporal granularity, explicit periodicity, and dynamic spatial heterogeneity. While DGCRN dynamically updates graph structures, it does so at high computational cost and without explicit periodic modeling; HEPHASTUS, in contrast, achieves better performance with lower parametric overhead through structured priors and efficient parameterization.

Finally, Models such as STID (Shao et al., 2022a) and STAEformer (Liu et al., 2023a) achieve competitive performance on certain datasets by decoupling from predefined graphs and encoding node and time identities into embeddings. However, their performance degrades notably on METR-LA and PEMS03, where traffic dynamics are more volatile and spatial dependencies more complex. This suggests that while static identity embeddings can capture fixed node characteristics, they are insufficient for modeling time-varying spatial relationships and transient temporal patterns. In contrast, HEPHASTUS explicitly models dynamic spatial heterogeneity through HSA and adaptive temporal scales through AMS-MoE, enabling it to respond to evolving traffic conditions more effectively.

In summary, the experimental results validate that **a unified, adaptive, and structured approach**, rather than incremental improvements to isolated components, is key to advancing traffic forecasting. HEPHASTUS's consistent gains across diverse settings demonstrate that its core innovations, adaptive multi-scale routing, periodic attention, and heterogeneous spatial modeling, are not only effective but necessary for capturing the intrinsic complexity of urban traffic systems.

## A.4 ABLATION STUDIES

To systematically evaluate the contribution of each component in **HEPHASTUS**, we conduct a detailed ablation study on four benchmark datasets: METR-LA, PEMS-BAY, PEMS04, and PEMS08.

Table 5: Statistics of datasets.

| Dataset | Samples | Variates | Frequency | Time Span | Graph |
|---------|---------|----------|-----------|-----------|-------|
| **METR-LA** | 34272 | 207 | 5 mins | 4 months | Yes |
| **PEMS-BAY** | 52116 | 325 | 5 mins | 6 months | Yes |
| **PEMS03** | 26208 | 358 | 5 mins | 3 months | Yes |
| **PEMS04** | 16992 | 307 | 5 mins | 2 months | Yes |
| **PEMS07** | 28224 | 883 | 5 mins | 4 months | Yes |
| **PEMS08** | 17856 | 170 | 5 mins | 2 months | Yes |

We assess the full model against five ablated variants, each designed to isolate the impact of a key architectural choice, under identical training and evaluation protocols. Results are reported in Table 3.

**Effect of Adaptive Multi-Scale Mixture of Experts (AMS-MoE).** We replace the AMS-MoE with a single fixed-scale encoder using a patch size of $P = 3$ (w/o AMS-MoE), thereby removing the model's ability to adaptively select temporal scales based on input dynamics. This variant suffers the largest performance degradation across all datasets, with MAE increasing by 4.5% on METR-LA and 6.9% on PEMS08. These results confirm that fixed-scale modeling limits the model's capacity to handle diverse temporal patterns, such as transient spikes during rush hours or smooth trends during off-peak periods, and underscore the necessity of input-adaptive scale decomposition.

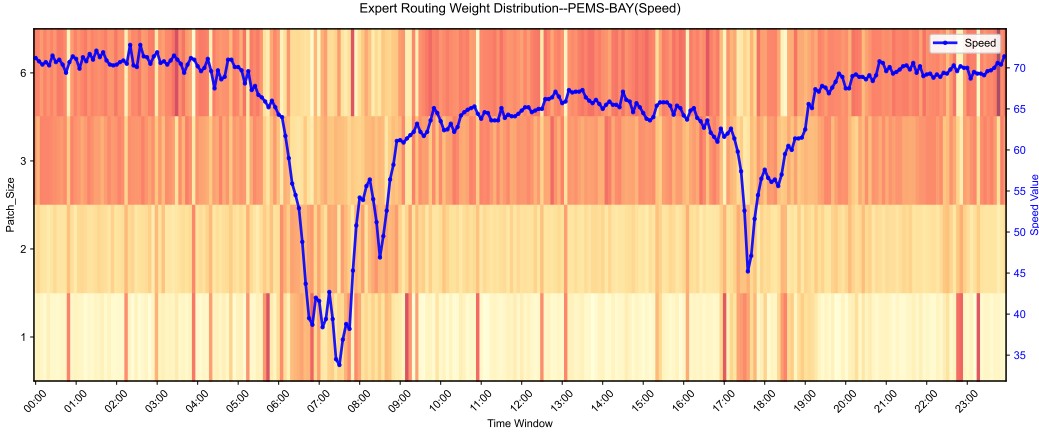

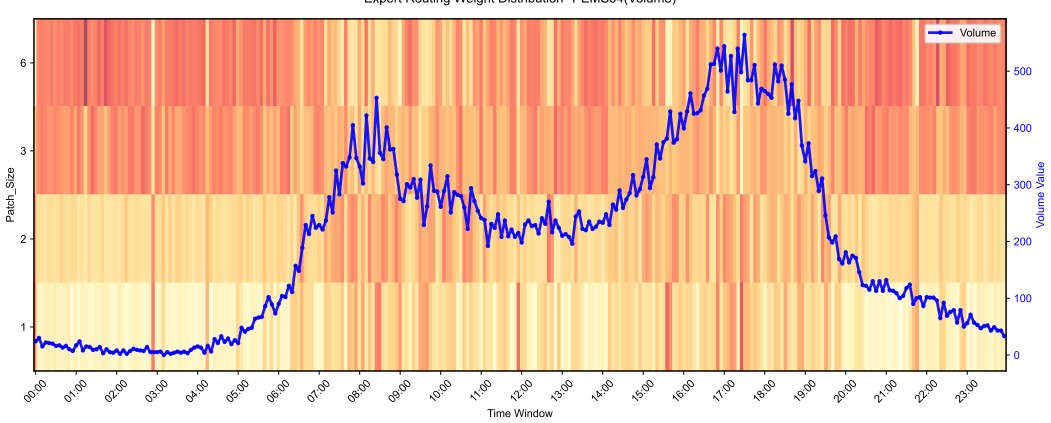

Figure 4: Case study on PEMS-BAY (top) and PEMS04 (bottom) datasets. Line plot: Ground truth traffic indicators (speed/volume). Heatmap: Corresponding expert selection probabilities from the AMS-MoE router, showing adaptive scale preference aligned with traffic volatility.

**Impact of Explicit Periodicity Modeling (PTA).** We ablate the Periodic Temporal Attention (PTA) by replacing it with standard temporal self-attention without periodic position encodings (w/o PTA). This removes the explicit modeling of daily and weekly cycles, forcing the model to implicitly learn periodic patterns from data. The resulting performance drop, 2.7% increase in MAE on METR-LA and 3.1% on PEMS08, demonstrates that explicit incorporation of temporal periodicity significantly enhances long-term dependency modeling, especially for multi-step forecasting where such regularities are critical.

**Role of Heterogeneous Spatial Dependencies (HSA).** We substitute the Heterogeneous Spatial Attention (HSA) with a shared global value projection $V = \mathcal{H}W_v$ (w/o HSA), eliminating node-specific transformations and the learnable pattern library $PL$. This assumes uniform spatial dependencies across all nodes, ignoring functional heterogeneity (e.g., arterial roads vs. local streets). The consistent degradation across metrics, especially in MAPE, which increases by up to 0.91%, indicates that modeling node-specific behaviors is crucial for accurate spatial representation. HSA's low-rank parametrization enables efficient personalization at minimal cost ($O(Nr + rCD)$), validating its design for large-scale networks.

**Necessity of Dynamic, Input-Dependent Routing.** We retain multiple parallel scale experts but disable the adaptive router, instead aggregating expert outputs using fixed uniform weights (Fixed-Scale MoE). This isolates the effect of dynamic routing by removing input-aware scale selection. Despite having the same model capacity, this variant underperforms the full model (+3.3% MAE on PEMS04), confirming that static fusion cannot match the flexibility of input-conditioned routing. The adaptive router enables context-sensitive allocation of expert capacity, improving alignment with real-world traffic dynamics.

**Value of Sparse Expert Activation.** We retain the full router architecture but remove the Top-$K$ sparsification, allowing all experts to contribute to the output (Noisy-Routing Only). While this maintains input-dependent routing, it eliminates sparsity, increasing computational load and reducing regularization. The resulting performance gap despite noise injection for exploration suggests that sparse activation not only improves efficiency but also enhances generalization by preventing over-reliance on any single expert. This supports the use of structured sparsity in MoE-based time series models.

In summary, each component of **HEPHASTUS** contributes meaningfully to its state-of-the-art performance. The ablation study validates that adaptive scale selection, explicit periodicity modeling, and heterogeneous spatial learning are not merely incremental improvements but essential mechanisms for capturing the intrinsic complexity of urban traffic. Together, they enable a flexible, efficient, and structured approach to spatio-temporal forecasting.

## A.5 PARAMETER SENSITIVITY ANALYSIS

We conduct a systematic ablation study to evaluate the sensitivity of **HEPHASTUS** to three key architectural hyperparameters—number of experts $M$, top-$K$ routing sparsity $K$, and pattern library rank $r$ on METR-LA and PEMS08 datasets. Results are summarized in Figure 3, revealing critical insights into the model's capacity, inductive bias, and scalability.

**Number of experts ($M$).** The choice of $M$ determines the diversity of temporal scales the model can adaptively exploit. As shown in the left subfigure of Figure 3, increasing $M$ from 2 to 4 consistently improves performance on both datasets, with METR-LA MAE decreasing from 3.41 to 3.36 and PEMS08 from 13.77 to 13.56. This indicates that decomposing temporal dynamics into four distinct, learnable pathways (e.g., sub-hourly, hourly, daily, and trend) enables more precise pattern specialization. However, further increasing $M$ to 5 yields only marginal gains (PEMS08: 13.53) and a slight degradation on METR-LA (3.37), suggesting that excessive expert specialization introduces routing instability and redundant computation without meaningful performance benefits. Thus, $M = 4$ provides an optimal trade-off between representational richness and training stability.

The number of experts, M, determines the granularity of temporal scale decomposition in AMS-MoE. Each expert corresponds to a specific patch size P, enabling the extraction of dynamic features over different time windows. The value of M is fundamentally constrained by the input sequence length H, for a given H, only a limited set of "meaningful and non-redundant" patch size combinations exist. Although increasing M (e.g., to M=5) theoretically allows finer scales (e.g., P=4, 12) to

be introduced, our experiments (Fig. 4) show diminishing performance gains and, in some datasets (e.g., METR-LA), even a slight performance drop, along with significantly increased computational cost (in terms of parameter count and training/inference time). Thus, M=4 represents the optimal balance between expressive capacity and practical efficiency under H=12. For longer H (e.g., H=24 or 48), M can be increased to 5 or 6 to include longer-period scales such as P=8, 12, and 24.

**Top-$K$ sparsity ($K$).** The routing mechanism's sparsity level critically influences both generalization and expert collaboration. We observe a clear sweet spot at $K = 2$: on METR-LA, MAE reaches 3.36, a 1.8% improvement over $K = 1$ (3.42), while PEMS08 improves from 13.75 ($K = 1$) to 13.56. This confirms that restricting routing to a single expert ($K = 1$) creates a bottleneck, limiting contextual integration across temporal behaviors. In contrast, $K = 3$ and $K = 4$ degrade performance (METR-LA: 3.38 and 3.40; PEMS08: 13.71 and 13.62), with $K = 4$ effectively reverting to dense, unregularized routing. The decline indicates that excessive expert activation undermines the sparsity prior, leading to overfitting and reduced interpretability. Hence, $K = 2$ enforces a balanced, selective fusion strategy that promotes robustness.

The Top-K parameter K controls the number of experts activated per input sequence during inference. It is not a fixed sparsity level, but rather a dynamic "decision granularity" that adapts to the information density of the input. K primarily depends on the input length H and scene complexity: for smaller H (e.g., H=6), where fewer temporal scales exist and information density is low, K=1 may suffice; for larger H (e.g., H=24), which contains richer scale dependencies, K=3 or 4 better facilitates expert collaboration. In our setting (H=12), K=2 ensures cross-scale interaction while avoiding system overload. Traffic data complexity varies significantly: for simple scenarios (e.g., highways, low volatility), K=1 or 2 is sufficient; for complex scenarios (e.g., urban signalized intersections, high volatility), more expert collaboration is required, making K=3 more appropriate. Our experiments confirm that K=2 is optimal for highway or expressway datasets (METR-LA, PEMS07, PEMS08). Moreover, larger K incurs higher inference cost; thus, for high-complexity or long-H scenarios, K can be increased to 3 or 4, while for edge deployment or low-complexity scenarios, K can be reduced to 1 or 2.

**Pattern library rank ($r$).** The rank $r$ controls the expressiveness of the low-rank HSA module in capturing spatial heterogeneity. Performance improves steadily as $r$ increases from 4 to 8: METR-LA MAE drops from 3.39 to 3.36, and PEMS08 from 13.73 to 13.55, reflecting enhanced modeling of node-specific patterns. Further increasing $r$ to 16 yields only minor gains (METR-LA: 3.34, PEMS08: 13.56), while $r = 32$ leads to performance degradation (PEMS08: 13.62), likely due to overfitting on spatial configurations with limited training signals. These results validate that spatial dependencies exhibit strong low-rank structure, and a modest $r = 8$ suffices to capture dominant heterogeneous patterns efficiently.

The rank r governs the dimensionality of the spatial pattern embedding, used to capture dominant spatial modes in the road network (e.g., congestion corridors, arterial roads, regional clusters). Its selection primarily depends on the network scale and traffic scenario complexity: spatial complexity increases with the number of nodes N, larger networks require larger r. Complex scenarios demand richer representations: for highway or expressway scenarios, where traffic flow is relatively homogeneous, a smaller r is sufficient to model node-specific characteristics. In contrast, for urban traffic scenarios (e.g., signal-controlled intersections, mixed traffic), nodes exhibit stronger heterogeneity, requiring a higher r to match the dynamically complex spatial patterns. However, larger r also increases model parameter count and computational cost during training and inference; therefore, in practical applications, a balance must be struck between accuracy and efficiency.

In summary, **HEPHASTUS** exhibits consistent behavior across hyperparameter settings, with peak performance concentrated around moderate values. The results underscore the importance of adaptive multi-scale decomposition, sparse expert routing, and compact spatial modeling—design choices that jointly enable scalable, robust, and interpretable spatio-temporal forecasting.

## A.6 CASE STUDY

To gain deeper insights into the adaptive behavior of our proposed AMS-MoE module, we conduct a qualitative analysis of its routing patterns on representative traffic sequences from PEMS-BAY and PEMS04 datasets. Figure 4 illustrates the temporal evolution of traffic indicators (speed/volume)

alongside the corresponding expert selection probabilities generated by our dynamic router for two sample sequences.

**Observation 1: Scale-Sensitive Routing in Peak Hours.** During morning and evening rush hours, we observe that traffic indicators exhibit rapid fluctuations and high volatility due to congestion dynamics. Correspondingly, the AMS-MoE router demonstrates a strong preference for experts operating at smaller patch sizes, which are specialized in capturing fine-grained, short-term variations. This adaptive selection enables the model to focus on transient patterns such as sudden slowdowns, stop-and-go waves, and local anomalies that characterize peak-hour traffic conditions.

**Observation 2: Coarse-Grained Emphasis in Off-Peak Periods.** Conversely, during off-peak hours (e.g., midday or late night), traffic flows tend to be smoother and more predictable, with gradual trends dominating the temporal dynamics. In these intervals, the router shifts its emphasis toward experts with larger patch sizes, which excel at modeling long-term dependencies and macroscopic trends. This strategic allocation allows the model to leverage stable periodic patterns and reduce unnecessary attention to noise or minor fluctuations.

**Mechanism Interpretation.** The routing behavior aligns with the intrinsic multi-scale characteristics of urban traffic: high-frequency components dominate during congested periods requiring localized, high-resolution analysis, while low-frequency patterns prevail in stable conditions where broader contextual understanding is sufficient. This input-dependent specialization demonstrates that AMS-MoE effectively decomposes temporal complexity without manual intervention, validating its role as an adaptive feature extractor that responds to the underlying data distribution.

**Comparative Advantage.** Fixed-scale architectures, by contrast, would either oversmooth peak-hour details (if using large patches) or overfit to off-peak noise (if using small patches). Our dynamic routing mechanism circumvents this trade-off by continuously adjusting the scale emphasis according to real-time traffic characteristics, contributing to the consistent performance gains observed in Section 5.3.

These case studies confirm that HEPHASTUS not only achieves quantitative improvements but also embodies a semantically meaningful adaptation strategy that mirrors the multi-scale nature of traffic dynamics. The ability to autonomously prioritize relevant temporal resolutions underscores the value of input-conditioned modeling in complex spatio-temporal forecasting tasks.

## A.7 LARGE-SCALE DATASET EVALUATION

To validate the scalability and effectiveness of **HEPHASTUS** on large-scale traffic forecasting scenarios, we conduct comprehensive experiments on the recently proposed LargeST benchmark (Liu et al., 2023b). LargeST represents one of the largest publicly available traffic datasets, containing over 8,600 sensors across four major regions: San Diego (SD), Greater Bay Area (GBA), Greater Los Angeles (GLA), and California (CA). This evaluation specifically addresses the model's ability to handle massive sensor networks and complex spatial dependencies at scale.

We follow the standard evaluation protocol established in (Liu et al., 2023b), using the same data splits and preprocessing procedures. For fair comparison, we select three representative baselines that have demonstrated strong performance on large-scale settings: STID (Shao et al., 2022a), GWNet (Wu et al., 2019), and STWave (Fang et al., 2023). All models are evaluated across three prediction horizons (3, 6, and 12 steps) using MAE, RMSE, and MAPE metrics. Table 6 presents the detailed performance comparison across all four regions of LargeST. **HEPHASTUS** achieves consistent and significant improvements over all baselines in nearly all scenarios.The exceptional performance on LargeST highlights several advantages of our architectural choices for large-scale settings:

1. The **Adaptive Multi-Scale Mixture of Experts (AMS-MoE)** effectively handles the diverse temporal patterns present across thousands of sensors without manual scale selection, automatically allocating computational resources to the most relevant temporal resolutions.

2. The **Heterogeneous Spatial Attention (HSA)** mechanism's low-rank parameterization ($O(Nr + rCD)$) scales efficiently with network size, avoiding the quadratic complexity of full attention while maintaining expressive power for node-specific modeling.

Table 6: Large-scale traffic forecasting performance comparison.

| Datasets | Methods | Horizon 3 | | | Horizon 6 | | | Horizon 12 | | | Average | | |
|---|---|---|---|---|---|---|---|---|---|---|---|---|---|
| | | MAE | RMSE | MAPE (%) | MAE | RMSE | MAPE (%) | MAE | RMSE | MAPE (%) | MAE | RMSE | MAPE (%) |
| SD | STID | 15.15 | 25.29 | 9.82 | 17.95 | 30.39 | 11.93 | 21.82 | 38.63 | 15.09 | 17.86 | 31.00 | 11.94 |
| | GWNET | 15.24 | 25.13 | 9.86 | 17.74 | 29.51 | 11.70 | 21.56 | 36.82 | 15.13 | 17.74 | 29.62 | 11.88 |
| | STWave | 15.80 | 25.89 | 10.34 | 18.18 | 30.03 | 11.96 | 21.98 | 36.99 | 15.30 | 18.22 | 30.12 | 12.20 |
| | HEPHASTUS | 14.26 | 24.19 | 9.06 | 16.67 | 29.02 | 11.76 | 20.28 | 35.98 | 14.38 | 16.56 | 29.17 | 11.01 |
| GBA | STID | 17.36 | 29.39 | 13.28 | 20.45 | 34.51 | 16.03 | 24.38 | 41.33 | 19.90 | 20.22 | 34.61 | 15.91 |
| | GWNET | 17.85 | 29.12 | 13.92 | 21.11 | 33.69 | 17.79 | 25.58 | 40.19 | 23.48 | 20.91 | 33.41 | 17.66 |
| | STWave | 17.95 | 29.42 | 13.01 | 20.99 | 34.01 | 15.62 | 24.96 | 40.31 | 20.08 | 20.81 | 33.77 | 15.76 |
| | HEPHASTUS | 16.63 | 27.96 | 12.46 | 18.95 | 32.56 | 14.11 | 22.89 | 39.12 | 18.10 | 19.26 | 32.69 | 14.31 |
| GLA | STID | 16.54 | 27.73 | 10.00 | 19.98 | 34.23 | 12.38 | 24.29 | 42.50 | 16.02 | 19.76 | 34.56 | 12.41 |
| | GWNET | 17.28 | 27.68 | 10.18 | 21.31 | 33.70 | 13.02 | 26.99 | 42.51 | 17.64 | 21.20 | 33.58 | 13.18 |
| | STWave | 17.48 | 28.05 | 10.06 | 21.08 | 33.58 | 12.56 | 25.82 | 41.28 | 16.51 | 20.96 | 33.48 | 12.70 |
| | HEPHASTUS | 15.18 | 25.89 | 9.06 | 18.92 | 31.09 | 11.23 | 22.99 | 38.96 | 14.37 | 18.26 | 31.89 | 11.16 |
| CA | STID | 15.51 | 26.23 | 11.26 | 18.53 | 31.56 | 13.82 | 22.63 | 39.37 | 17.59 | 18.41 | 32.00 | 13.82 |
| | GWNET | 17.14 | 27.81 | 12.62 | 21.68 | 34.16 | 17.14 | 28.58 | 44.13 | 24.24 | 21.72 | 34.20 | 17.40 |
| | STWave | 16.77 | 26.98 | 12.20 | 18.97 | 30.69 | 14.40 | 25.36 | 38.77 | 19.01 | 19.69 | 31.58 | 14.58 |
| | HEPHASTUS | 14.28 | 24.89 | 10.18 | 17.09 | 29.46 | 12.38 | 20.93 | 36.15 | 15.87 | 17.05 | 29.83 | 12.31 |

3. The **Periodic Temporal Attention (PTA)** provides structured inductive biases that remain effective even when spatial dependencies become increasingly complex in large-scale networks.

These results conclusively demonstrate that **HEPHASTUS** not only excels on medium-scale benchmarks but also maintains state-of-the-art performance on truly large-scale traffic forecasting tasks, and establishing its practical utility for real-world deployment in complex urban environments.

## A.8 CROSS-DOMAIN EVALUATION ON CRIME PREDICTION

To regard the generalizability of our approach beyond traffic forecasting, we conduct extensive experiments on urban crime prediction, a fundamentally different spatio-temporal forecasting domain with distinct challenges including high sparsity, complex socio-economic factors, and multi-type crime dynamics. This evaluation validates whether the adaptive multi-scale modeling and heterogeneous spatial learning principles in **HEPHASTUS** transfer effectively to other real-world forecasting scenarios.

We evaluate on two real-world crime datasets from New York City (NYC) and Chicago (CHI) following established benchmarks (Xia et al., 2022; Li et al., 2022). These datasets contain crime event records across multiple categories (Burglary, Larceny, Robbery, Assault in NYC; Theft, Battery, Assault, Damage in CHI) with inherent spatial-temporal dependencies. We compare against three categories of baselines: (1) General STGNNs: GWNet (Wu et al., 2019), AGCRN (Bai et al., 2020), MT-GNN (Wu et al., 2020); (2) Crime-specific methods: DeepCrime (Huang et al., 2018), STSHN (Xia et al., 2022), STHSL (Li et al., 2022). Table 7 presents the comprehensive results across all crime categories and both cities. **HEPHASTUS** achieves remarkable performance improvements, consistently outperforming all specialized crime prediction methods and general STGNNs by significant margins:The exceptional performance on crime prediction reveals several transferable strengths of our framework:

1. The **Adaptive Multi-Scale Mixture of Experts (AMS-MoE)** effectively captures the multi-resolution temporal patterns in crime data, which exhibit both short-term bursts (e.g., weekend spikes) and long-term seasonal trends (e.g., seasonal variations in crime rates), without requiring domain-specific modifications.

2. The **Heterogeneous Spatial Attention (HSA)** mechanism naturally adapts to the heterogeneous nature of urban crime distributions, where different regions (e.g., commercial districts vs. residential areas) exhibit distinct crime patterns that require personalized modeling approaches.

3. The **Periodic Temporal Attention (PTA)** successfully encodes crime-specific periodicities, such as weekly cycles (weekday vs. weekend variations), and seasonal trends, demonstrating the generalizability of explicit periodicity modeling.

Table 7: Overall performance of urban crime prediction on NYC and CHI dataset

| Model | New York City | | | | | | | | Chicago | | | | | | | |
|---|---|---|---|---|---|---|---|---|---|---|---|---|---|---|---|---|
| | Burglary | | Larceny | | Robbery | | Assault | | Theft | | Battery | | Assault | | Damage | |
| | MAE | MAPE | MAE | MAPE | MAE | MAPE | MAE | MAPE | MAE | MAPE | MAE | MAPE | MAE | MAPE | MAE | MAPE |
| GWNet | 0.7993 | 0.5235 | 1.0493 | 0.5405 | 0.8681 | 0.5351 | 0.8866 | 0.5646 | 1.3211 | 0.5502 | 1.1331 | 0.5503 | 0.7493 | 0.4580 | 0.8584 | 0.4850 |
| AGCRN | 0.8260 | 0.5397 | 1.0499 | 0.5404 | 0.9013 | 0.5383 | 0.9063 | 0.5519 | 1.3281 | 0.5304 | 1.1432 | 0.5697 | 0.7669 | 0.4612 | 0.8712 | 0.4859 |
| MTGNN | 0.8329 | 0.5439 | 1.0473 | 0.5330 | 0.8759 | 0.5457 | 0.9090 | 0.5714 | 1.3054 | 0.5378 | 1.1307 | 0.5597 | 0.7571 | 0.4572 | 0.8667 | 0.4859 |
| DeepCrime | 0.8227 | 0.5508 | 1.0618 | 0.5351 | 0.8841 | 0.5537 | 0.9222 | 0.5677 | 1.3391 | 0.5430 | 1.1290 | 0.5389 | 0.7737 | 0.4616 | 0.9096 | 0.4960 |
| STSHN | 0.8012 | 0.5198 | 1.0431 | 0.5291 | 0.8717 | 0.5362 | 0.9169 | 0.5682 | 1.3231 | 0.5310 | 1.1348 | 0.5544 | 0.7758 | 0.4574 | 0.8741 | 0.4747 |
| STHSL | 0.7329 | 0.4788 | 1.0316 | 0.5040 | 0.7912 | 0.4595 | 0.8484 | 0.5029 | 1.2952 | 0.4929 | 1.1016 | 0.5231 | 0.6665 | 0.3996 | 0.8446 | 0.4644 |
| HEPHASTUS | **0.5634** | **0.2758** | **0.9124** | **0.4416** | **0.7254** | **0.3141** | **0.6929** | **0.4051** | **1.0625** | **0.4176** | **0.9628** | **0.4418** | **0.5863** | **0.3215** | **0.6729** | **0.3507** |

These results demonstrate that **HEPHASTUS** is not merely a specialized traffic forecasting model but represents a general framework for spatio-temporal forecasting that transfers effectively across domains. The consistent superiority over domain-specific methods (ST-SHN, ST-HSL) that incorporate sophisticated crime-specific inductive biases is particularly noteworthy, suggesting that our adaptive, structured approach provides a more fundamental solution to spatio-temporal modeling challenges.

## A.9 THEORETICAL COMPLEXITY ANALYSIS

The computational complexity of the HEPHASTUS model primarily stems from its three core modules: AMS-MoE, PTA, and HSA. The AMS-MoE module is responsible for adaptive multi-scale temporal modeling, with a computational complexity of $O(NHCD)$. The PTA module explicitly models daily/weekly periodicity, and its complexity is dominated by temporal attention computation, which has a complexity of $O(NH^2D)$. The HSA module models dynamic spatial heterogeneity, and its complexity primarily arises from the computation of node-specific transformation matrices and spatial attention. Specifically, the parameter count for the node embedding $S_e \in \mathbb{R}^{N \times r}$ and pattern library $PL \in \mathbb{R}^{r \times C \times D}$ is $O(Nr + rCD)$, and the computational cost is $O(NrCD)$. Since $r$ is a small constant, this can be approximated as $O(NCD)$. The spatial attention computation has a complexity of $O(N^2HD)$. In conventional Transformer spatial attention modules, modeling node-specific characteristics typically requires assigning independent feature transformation matrices to each node, resulting in a parameter count of $O(NCD)$. This leads to high GPU memory consumption for optimizer states (e.g., momentum and variance in Adam) and substantial computational overhead for gradient calculation and parameter updates. In contrast, HSA achieves comparable or superior performance using fewer parameters via low-rank parameterization, thereby reducing the computational and communication overhead associated with parameter updates—especially beneficial in distributed training scenarios.

The overall complexity of HEPHASTUS is the sum of the complexities of the AMS-MoE, PTA, and HSA modules. Since these modules are executed sequentially, the total complexity is determined by the maximum complexity among them: $O(\max(NHCD, NH^2D, N^2HD))$. In practice, the dominant term depends on the relative sizes of $H$ and $N$, if $H$ is large (e.g., long-sequence prediction), the $O(NH^2D)$ complexity of PTA may dominate; if $N$ is large (e.g., large-scale road networks), the $O(N^2HD)$ complexity of HSA may dominate. The $O(NHCD)$ complexity of AMS-MoE is typically linear and does not dominate. Regarding the hyperparameter $M$ (number of experts), since HEPHASTUS employs a MoE architecture, $M$ affects only model capacity and expert load balancing during training, not inference complexity (as only $K$ experts are activated). For $K$ (number of activated experts), since $K$ is a constant (e.g., $K = 2$), its impact is $O(1)$ and does not affect the asymptotic complexity trend. As for $r$ (rank of the pattern library), it influences only the parameter count of HSA ($O(Nr)$) and a portion of its computational cost ($O(NrCD)$), but since $r$ is a small constant, its impact on complexity as $N$ and $H$ grow is negligible. While the HSA module maintains the same computational complexity as standard spatial attention (i.e., $O(N^2HD)$), its highly expressive, parameter-efficient design enables us to achieve superior performance with a smaller model capacity (e.g., fewer layers, smaller hidden dimensions). in turn, leads to reduced overall computational cost during both training and inference. For instance, as shown in Table 4, HEPHASTUS achieves state-of-the-art performance with only 716K parameters, which is significantly fewer than many Transformer-based baselines (e.g., 1263K for PDFormer), while also demonstrating faster inference speed and lower GPU memory usage.

In summary, the computational complexity of HEPHASTUS scales polynomially with $N$ and $H$, and the key hyperparameters $M$, $K$, and $r$ have limited impact on complexity. This demonstrates the model's strong scalability when handling large-scale traffic networks (large $N$) and long-sequence prediction tasks (large $H$). Its efficient parameter design (e.g., the low-rank structure of HSA and the sparse activation of AMS-MoE) enables the use of a smaller model capacity while maintaining high performance, thereby achieving lower computational cost in practice.

