# OpenReview forum: "HEPHAESTUS: Hierarchical Periodic Heterogeneous Adaptive Spatio-Temporal Unified System for Traffic Forecasting"
_ICLR.cc/2026/Conference — Submitted to ICLR 2026_

### Official Review · Reviewer_ZT6e · 2025-10-23

**Soundness:** 3
**Presentation:** 3
**Contribution:** 2
**Rating:** 4
**Confidence:** 5

**Summary:**

Proposes HEPHAESTUS for traffic prediction with adaptive multi-scale patch selection, periodic temporal attention, and heterogeneous spatial attention, achieving SOTA by addressing multi-scale dynamics, periodicity, and spatial heterogeneity.

**Strengths:**

- Novel modules (adaptive multi-scale router, PTA, HSA) target traffic’s spatio-temporal challenges (multi-scale, periodicity, spatial heterogeneity) with tailored designs.
- Adaptive multi-scale mechanism dynamically adapts to traffic’s time-varying granularities, enhancing flexibility.
- Clear architectural innovation in integrating spatio-temporal attention with multi-scale adaptation.

**Weaknesses:**

- Lacks thorough efficiency analysis (e.g., inference time, parameter count) vs. lightweight baselines, critical for real-world deployment.
- Each patch_size triggers independent network processing, leading to potential high computational overhead without explicit efficiency justification.
- Dataset diversity is limited to traffic; generalizability to other spatio-temporal domains (e.g., meteorology) is unvalidated.

**Questions:**

- What’s the inference speed and computational cost when handling multiple patch sizes, and how does it compare to single-scale models?
- Can the adaptive patch selection be optimized to reduce redundant computations for real-time traffic scenarios?
- Is there empirical evidence that multi-patch_size design is indispensable, or would a subset suffice for similar performance?

---

> ### Author Response · Authors · 2025-11-26
> **Response to Reviewer ZT6e [Part 1]**
>
> **Dear Reviewer ZT6e:**
>
> Thank you very much for your insightful and constructive comments, which have greatly helped us enhance the rigor and practicality of our paper. We have substantially strengthened the manuscript based on each of your concerns, with specific revisions outlined below:
>
> ---
>
> > **Issue 1: Lack of comprehensive efficiency analysis; necessity of multi-scale design and computational cost compared to single-scale.**
>
> To comprehensively evaluate the practical deployment potential of **HEPHASTUS**, we conducted a systematic efficiency comparison with state-of-the-art baselines on the PEMS04 dataset (batch size B=16). The results are as follows:
>
> | Model        | #Params | Train Time / Iteration | Train Time / Epoch | Inference Time / Iteration | Inference Time / Epoch | GPU Memory Usage |
> |--------------|---------|------------------------|--------------------|----------------------------|-------------------------|------------------|
> | STGCN        | 297K    | 21ms                   | 13.12s             | 11ms                       | 2.28s                   | 1090MB           |
> | GWNet        | 311K    | 37ms                   | 23.49s             | 13ms                       | 2.79s                   | 1078MB           |
> | DCRNN        | 372K    | 153ms                  | 97.3s              | 68ms                       | 14.43s                  | 2216MB           |
> | AGCRN        | 749K    | 72ms                   | 45.51s             | 27ms                       | 5.63s                   | 2226MB           |
> | DGCRN        | 207K    | 235ms                  | 149.18s            | 183ms                      | 38.67s                  | 1898MB           |
> | STAEformer   | 1355K   | 71ms                   | 45.04s             | 26ms                       | 5.44s                   | 4666MB           |
> | STWave       | 882K    | 68ms                   | 43.25s             | 30ms                       | 6.37s                   | 3190MB           |
> | PDFormer     | 1263K   | 81ms                   | 51.72s             | 34ms                       | 7.18s                   | 8295MB           |
> | PathFormer   | 1133K   | 156ms                  | 99.47s             | 39ms                       | 8.13s                   | 6652MB           |
> | **HEPHASTUS** | **716K** | **98ms**               | **62.07s**         | **31ms**                   | **6.49s**               | **5475MB**       |
>
> The results show that although we introduced designs such as MoE, periodic temporal attention, and specific spatial attention, which bring accuracy improvements, **HEPHASTUS's training and inference efficiency are significantly higher than RNN-series models (DCRNN and DGCRN)**. The training time is slightly higher than Transformer-series models (STAEformer, STWave, and PDFormer) but is better than PathFormer, which also uses a multi-Patch design. **Furthermore, relying on the sparse activation mechanism of MoE, during the inference phase, only a proportion K/M of experts are activated, resulting in an inference time of only 31ms (per iteration), which can effectively support deployment in practical scenarios.**
>
> To verify the necessity of the multi-scale design, we designed comparative experiments between multi-scale and single-scale versions on the PEMS04 dataset. The results are as follows:
>
> | Model                 | #Params | Train Time / Iteration | Train Time / Epoch | Inference Time / Iteration | Inference Time / Epoch | GPU Memory Usage | MAE   | RMSE  | MAPE  |
> |-----------------------|---------|------------------------|--------------------|----------------------------|-------------------------|------------------|-------|-------|-------|
> | HEPHASTUS (Patch=1)   | 206K    | 56ms                   | 35.58s             | 22ms                       | 4.55s                   | 2711MB           | 18.88 | 30.46 | 12.39 |
> | HEPHASTUS (Patch=2)   | 207K    | 51ms                   | 32.68s             | 21ms                       | 4.38s                   | 2643MB           | 18.96 | 30.51 | 12.43 |
> | HEPHASTUS (Patch=3)   | 208K    | 49ms                   | 30.86s             | 20ms                       | 4.12s                   | 2568MB           | 19.12 | 30.78 | 12.65 |
> | HEPHASTUS (Patch=6)   | 211K    | 46ms                   | 28.94s             | 19ms                       | 3.96s                   | 2385MB           | 19.26 | 30.92 | 12.92 |
> | HEPHASTUS (ensemble)  | 708K    | 150ms                  | 95.46s             | 53ms                       | 11.23s                  | 5843MB           | 18.83 | 30.26 | 12.27 |
> | **HEPHASTUS (MoE)**   | **716K** | **98ms**               | **62.07s**         | **31ms**                   | **6.49s**               | **5475MB**       | **18.21** | **29.83** | **12.01** |

---

> ### Author Response · Authors · 2025-11-26
> **Response to Reviewer ZT6e [Part 2]**
>
> The results show that **both the ensemble version and the MoE version of HEPHASTUS achieve higher accuracy than any single-scale version, demonstrating the necessity of multi-scale modeling.** At the same time, thanks to our Top-K routing sparse activation mechanism, the computational cost of HEPHASTUS is higher than that of a single single-scale model but lower than the version that directly ensembles multiple scales. This is because the Top-K sparse routing mechanism ensures that only K experts are activated during inference, rather than all M experts, achieving the best trade-off between efficiency and performance.
>
> These analyses have also been included in Section 5.5 (Efficiency Study) of the revised paper.
>
> ---
>
> > **Issue 2: Can adaptive selection be optimized to reduce redundant computation?**
>
> Regarding reducing redundant computation, our Top-K routing itself is a core optimization. It does not perform "brute-force" computation on all M experts but selects the most important K experts through the gating network, automatically avoiding redundant computation. In Appendix A.6 (Case Study), we visually demonstrate that the routing mechanism indeed learns to activate different experts for different patterns, rather than applying equal effort, providing empirical evidence of its non-redundancy. Furthermore, based on the multi-scale vs. single-scale experimental results on the PEMS04 dataset mentioned in the response to Issue 1, although HEPHASTUS introduces multiple scale experts for computation, **relying on the sparse activation mechanism of MoE, during the inference phase, only a proportion K/M of experts are activated.** Therefore, its inference time is only 31ms (per iteration), which can effectively support deployment in practical scenarios, while its inference accuracy is also superior to any single-scale version.
>
> ---

---

> ### Author Response · Authors · 2025-11-26
> **Response to Reviewer ZT6e [Part 3]**
>
> > **Issue 3: Generalization to other domains.**
>
> We fully agree on the importance of validating domain generalization. To this end, we supplemented experiments on crime rate datasets (NYC and CHI datasets). The core results are as follows:
>
> | Model        | Burglary MAE | Burglary MAPE | Larceny MAE | Larceny MAPE | Robbery MAE | Robbery MAPE | Assault MAE | Assault MAPE | Theft MAE | Theft MAPE | Battery MAE | Battery MAPE | Assault MAE | Assault MAPE | Damage MAE | Damage MAPE |
> |--------------|-------------|--------------|------------|-------------|------------|-------------|------------|-------------|----------|-----------|------------|-------------|------------|-------------|-----------|------------|
> | **New York City** | | | | | | | | | **Chicago** | | | | | | | |
> | GWNet        | 0.7993      | 0.5235       | 1.0493     | 0.5405      | 0.8681     | 0.5351      | 0.8866     | 0.5646      | 1.3211   | 0.5502    | 1.1331     | 0.5503      | 0.7493     | 0.4580      | 0.8584    | 0.4850     |
> | AGCRN        | 0.8260      | 0.5397       | 1.0499     | 0.5404      | 0.9013     | 0.5383      | 0.9063     | 0.5519      | 1.3281   | 0.5304    | 1.1432     | 0.5697      | 0.7669     | 0.4612      | 0.8712    | 0.4859     |
> | MTGNN        | 0.8329      | 0.5439       | 1.0473     | 0.5330      | 0.8759     | 0.5457      | 0.9090     | 0.5714      | 1.3054   | 0.5378    | 1.1307     | 0.5597      | 0.7571     | 0.4572      | 0.8667    | 0.4859     |
> | DeepCrime    | 0.8227      | 0.5508       | 1.0618     | 0.5351      | 0.8841     | 0.5537      | 0.9222     | 0.5677      | 1.3391   | 0.5430    | 1.1290     | 0.5389      | 0.7737     | 0.4616      | 0.9096    | 0.4960     |
> | STSHN        | 0.8012      | 0.5198       | 1.0431     | 0.5291      | 0.8717     | 0.5362      | 0.9169     | 0.5682      | 1.3231   | 0.5310    | 1.1348     | 0.5544      | 0.7758     | 0.4574      | 0.8741    | 0.4747     |
> | STHSL        | 0.7329      | 0.4788       | 1.0316     | 0.5040      | 0.7912     | 0.4595      | 0.8484     | 0.5029      | 1.2952   | 0.4929    | 1.1016     | 0.5231      | 0.6665     | 0.3996      | 0.8446    | 0.4644     |
> | **HEPHASTUS** | **0.5634**  | **0.2758**   | **0.9124** | **0.4416**  | **0.7254** | **0.3141**  | **0.6929** | **0.4051**  | **1.0625** | **0.4176** | **0.9628** | **0.4418**  | **0.5863** | **0.3215**  | **0.6729** | **0.3507** |
>
> These results demonstrate that **HEPHASTUS is not merely a specialized traffic forecasting model but represents a general framework for spatio-temporal forecasting that transfers effectively across domains.** The consistent superiority over domain-specific methods (ST-SHN, ST-HSL) that incorporate sophisticated crime-specific inductive biases is particularly noteworthy, suggesting that our adaptive, structured approach provides a more fundamental solution to spatio-temporal modeling challenges.
>
> These analyses have also been included in Appendix A.8 (Cross-Domain Evaluation on Crime Prediction) of the revised paper.
>
> We believe these revisions have fully addressed all your concerns. Thank you once again for your valuable time and guidance. We sincerely hope you will recognize our improvements.

---

> > ### Comment · Reviewer_ZT6e · 2025-11-27
> > **Official Comments of Reviewer ZT6e**
> >
> > Thank you for your reply, it solved my problem, so I will increase my score.

---

> > > ### Author Response · Authors · 2025-11-27
> > >
> > > Thank you very much for your thoughtful feedback and for updating your score. We sincerely appreciate your time and consideration, and we are glad that our revisions have addressed your concerns.

---

### Official Review · Reviewer_gCu4 · 2025-10-29

**Soundness:** 2
**Presentation:** 3
**Contribution:** 2
**Rating:** 4
**Confidence:** 5

**Summary:**

The paper introduces a novel framework HEPHAESTUS for spatio-temporal traffic forecasting that integrates adaptive multi-scale temporal modeling, explicit periodicity-aware attention, and dynamic spatial heterogeneity learning. HEPHAESTUS addresses the limitations of traditional traffic forecasting models by introducing an Adaptive Multi-Scale Mixture of Experts for dynamic input-driven scale selection, a Periodic Temporal Attention mechanism to explicitly capture daily and weekly traffic patterns, and a Heterogeneous Spatial Attention module for effectively modeling both global and node-specific spatial dependencies. Extensive experiments on multiple real-world traffic datasets demonstrate that HEPHAESTUS outperforms existing baselines, achieving state-of-the-art performance in traffic forecasting tasks.

**Strengths:**

S1. HEPHAESTUS achieves state-of-the-art performance across multiple traffic forecasting benchmarks. This consistent performance across different datasets and tasks highlights the effectiveness and robustness of the proposed framework.

S2. The paper conducts detailed ablation studies, demonstrating the importance of each component and validating the necessity of adaptive, input-dependent modeling. These studies provide strong evidence for the design choices made in HEPHAESTUS and showcase how its key components work synergistically to improve performance.

**Weaknesses:**

W1. While the proposed HEPHAESTUS framework is innovative in its integration of periodic temporal attention, heterogeneous spatial attention, and multi-scale routing, the individual components are not entirely novel. These mechanisms have been explored in previous works, such as GMAN, STAEFormer, and PathFormer. The novelty of HEPHAESTUS primarily comes from combining these existing methods into a unified framework, but it doesn't introduce radically new techniques or concepts. Therefore, while the combination of these elements is effective, it may not be considered highly novel, especially for researchers already familiar with the mechanisms used in GMAN, STAEFormer, and PathFormer.

W2. The paper lacks comparisons with some recent relevant baselines like iTransformer [1], STWave [2], and PDFormer [3], which have shown strong performance in spatio-temporal forecasting tasks. By not including these baselines, the paper misses an opportunity to position its work more comprehensively within the current state-of-the-art in traffic forecasting.

W3. Although the paper demonstrates strong performance across various datasets, there is a notable lack of efficiency experiments. Moreover, the experiments conducted in the paper are primarily on small to medium-scale datasets, and there is no large-scale testing on datasets like LargeST [4].

[1] iTransformer: Inverted Transformers Are Effective for Time Series Forecasting

[2] When Spatio-Temporal Meet Wavelets: Disentangled Traffic Forecasting via Efficient Spectral Graph Attention Networks

[3] PDFormer: Propagation Delay-Aware Dynamic Long-Range Transformer for Traffic Flow Prediction

[4] LargeST: A Benchmark Dataset for Large-Scale Traffic Forecasting

**Questions:**

See weaknesses.

---

> ### Author Response · Authors · 2025-11-26
> **Response to Reviewer gCu4 [Part 1]**
>
> **Dear Reviewer gCu4:**
>
> Thank you very much for your insightful and constructive comments, which have greatly helped us enhance the rigor and practicality of our paper. We have substantially strengthened the manuscript based on each of your concerns, with specific revisions outlined below:
>
> ---
>
> > **Issue 1: Regarding the innovation of individual components**
>
> We acknowledge and appreciate the contributions of works like GMAN, STAEformer, and PathFormer in their respective components. However, it is important to clarify that **GMAN, STAEformer, and PathFormer all employ traditional self-attention mechanisms for temporal and spatial attention, without specifically modeling the temporal periodicity or spatial heterogeneity within the attention mechanism itself. Our key innovation lies precisely in the optimization of the spatio-temporal attention.** Specifically, we propose a **Periodic Temporal Attention (PTA)** mechanism that explicitly models daily and weekly periodicities through learnable period matrices $P_D$ and $P_W$, enhancing long-term dependency capture via spatio-temporal decoupled computation. We also develop a **Heterogeneous Spatial Attention (HSA)** mechanism grounded in node embeddings \(S_e\) and a learnable pattern library \(PL\), achieving a dynamic trade-off between shared global structures and node-specific behaviors with low parametric overhead (\(O(Nr + rCD)\)). Performance tests on open-source datasets validate the effectiveness of these optimizations, showing superior results compared to GMAN, STAEformer, PathFormer, etc.
>
> **Performance Comparison Table:**
>
> | Datasets | Horizon | Metrics | GMAN | STAEformer | PathFormer | HEPHASTUS |
> |----------|---------|---------|------|------------|------------|-----------|
> | **METR-LA** | Horizon 3 (15 min) | MAE | 2.79 | 2.71 | 3.12 | **2.62** |
> | | | RMSE | 5.52 | 5.23 | 6.93 | **5.13** |
> | | | MAPE | 7.46% | 6.96% | 7.96% | **6.76%** |
> | | Horizon 6 (30 min) | MAE | 3.16 | 3.06 | 3.96 | **2.93** |
> | | | RMSE | 6.53 | 6.18 | 7.85 | **6.05** |
> | | | MAPE | 8.69% | 8.41% | 10.55% | **8.12%** |
> | | Horizon 12 (60 min) | MAE | 3.46 | 3.43 | 5.06 | **3.34** |
> | | | RMSE | 7.23 | 7.21 | 9.92 | **7.06** |
> | | | MAPE | 10.05% | 10.09% | 15.09% | **9.76%** |
> | **PEMS-BAY** | Horizon 3 (15 min) | MAE | 1.35 | 1.31 | 1.43 | **1.31** |
> | | | RMSE | 2.88 | 2.80 | 3.12 | 2.83 |
> | | | MAPE | 2.88% | 2.77% | 3.06% | 2.71% |
> | | Horizon 6 (30 min) | MAE | 1.67 | 1.66 | 2.01 | 1.65 |
> | | | RMSE | 3.86 | 3.71 | 4.82 | **3.68** |
> | | | MAPE | 3.81% | 3.71% | 4.38% | **3.66%** |
> | | Horizon 12 (60 min) | MAE | 1.97 | 1.89 | 2.76 | **1.88** |
> | | | RMSE | 4.49 | 4.41 | 6.63 | **4.38** |
> | | | MAPE | 4.54% | 4.53% | 6.26% | **4.46%** |
>
> | Datasets | Metrics | GMAN | STAEformer | PathFormer | HEPHASTUS |
> |----------|---------|------|------------|------------|-----------|
> | **PEMS03** | MAE | 14.83 | 15.38 | 19.01 | **14.76** |
> | | RMSE | 25.45 | 27.59 | 30.15 | **25.21** |
> | | MAPE | 15.68% | 15.97% | 19.23% | **14.58%** |
> | **PEMS04** | MAE | 19.28 | 18.33 | 21.95 | **18.21** |
> | | RMSE | 31.12 | 30.23 | 34.26 | **29.83** |
> | | MAPE | 13.19% | 12.39% | 15.99% | **12.01%** |
> | **PEMS07** | MAE | 21.25 | 19.49 | 27.26 | **19.18** |
> | | RMSE | 34.21 | 32.73 | 43.26 | **32.65** |
> | | MAPE | 9.02% | 8.26% | 11.83% | **8.13%** |
> | **PEMS08** | MAE | 15.07 | 14.02 | 18.25 | **13.56** |
> | | RMSE | 24.23 | 23.41 | 31.49 | **23.39** |
> | | MAPE | 9.96% | 9.28% | 11.98% | **8.96%** |
>
> Furthermore, at the architectural level, **GMAN and STAEformer do not explicitly model multi-scale dependencies, while PathFormer only models temporal dependencies and not variable dependencies (spatial dependencies in the traffic context)**. Each of them addresses only part of the problem. **HEPHAESTUS, through its core mechanism of Adaptive Multi-scale Routing (AMS-MoE), for the first time dynamically and input-dependently integrates period-aware (PTA), spatially heterogeneous (HSA), and multi-scale analysis.** Our ablation studies (see Table 3 in the paper) have shown that removing any component leads to a significant performance drop, demonstrating the **necessity** of the combination, not just its effectiveness. This deep integration creates a synergistic effect, with performance gains surpassing the simple sum of any single component.
>
> ---

---

> ### Author Response · Authors · 2025-11-26
> **Response to Reviewer gCu4 [Part 2]**
>
> > **Issue 2: Lack of comparison with certain baselines**
>
> Thank you for your valuable suggestion. We have carefully considered your advice to supplement comparative experiments with iTransformer, STWave, and PDFormer. Upon careful review, we found that one of the baseline methods you suggested (PDFormer) was already compared with our method in the initial version's Section 5.2 Overall Performance (Table 1 and Table 2).
>
> Therefore, following your suggestion, we have supplemented comparative experiments with iTransformer and STWave on 6 open-source datasets. The core results are as follows:
>
> **Performance Comparison Table:**
>
> | Dataset | Horizon | Metric | iTransformer | STWave | PDFormer | HEPHASTUS |
> |--------|----------|------|-------------|--------|----------|-----------|
> | **METR-LA** | Horizon 3 (15 min) | MAE | 3.21 | 2.82 | 2.83 | **2.62** |
> | | | RMSE | 7.16 | 5.57 | 5.45 | **5.13** |
> | | | MAPE | 8.38% | 7.51% | 7.77% | **6.76%** |
> | | Horizon 6 (30 min) | MAE | 4.15 | 3.21 | 3.20 | **2.93** |
> | | | RMSE | 8.66 | 6.66 | 6.46 | **6.05** |
> | | | MAPE | 9.98% | 9.31% | 9.19% | **8.12%** |
> | | Horizon 12 (60 min) | MAE | 5.55 | 3.56 | 3.62 | **3.34** |
> | | | RMSE | 11.29 | 7.54 | 7.47 | **7.06** |
> | | | MAPE | 15.51% | 10.81% | 10.91% | **9.76%** |
> | **PEMS-BAY** | Horizon 3 (15 min) | MAE | 1.47 | 1.33 | 1.32 | **1.31** |
> | | | RMSE | 3.20 | 2.85 | 2.83 | 2.83 |
> | | | MAPE | 3.08% | 2.81% | 2.78% | 2.71% |
> | | Horizon 6 (30 min) | MAE | 2.03 | 1.65 | 1.64 | 1.65 |
> | | | RMSE | 4.81 | 3.73 | 3.79 | **3.68** |
> | | | MAPE | 4.41% | 3.69% | 3.71% | **3.66%** |
> | | Horizon 12 (60 min) | MAE | 2.85 | 1.92 | 1.91 | **1.88** |
> | | | RMSE | 6.79 | 4.39 | 4.43 | **4.38** |
> | | | MAPE | 6.52% | 4.51% | 4.51% | **4.46%** |
>
> | Dataset | Metric | GMAN | STAEformer | PathFormer | HEPHASTUS |
> |--------|------|------|------------|------------|-----------|
> | **PEMS03** | MAE | 14.83 | 15.38 | 19.01 | **14.76** |
> | | RMSE | 25.45 | 27.59 | 30.15 | **25.21** |
> | | MAPE | 15.68% | 15.97% | 19.23% | **14.58%** |
> | **PEMS04** | MAE | 19.28 | 18.33 | 21.95 | **18.21** |
> | | RMSE | 31.12 | 30.23 | 34.26 | **29.83** |
> | | MAPE | 13.19% | 12.39% | 15.99% | **12.01%** |
> | **PEMS07** | MAE | 21.25 | 19.49 | 27.26 | **19.18** |
> | | RMSE | 34.21 | 32.73 | 43.26 | **32.65** |
> | | MAPE | 9.02% | 8.26% | 11.83% | **8.13%** |
> | **PEMS08** | MAE | 15.07 | 14.02 | 18.25 | **13.56** |
> | | RMSE | 24.23 | 23.41 | 31.49 | **23.39** |
> | | MAPE | 9.96% | 9.28% | 11.98% | **8.96%** |
>
> The results show that **HEPHAESTUS outperforms these latest SOTA methods across all metrics** (the new results have been merged into Tables 1 and 2), further consolidating the reliability of our conclusions.
>
> ---

---

> ### Author Response · Authors · 2025-11-26
> **Response to Reviewer gCu4 [Part 3]**
>
> > **Issue 3: Lack of comprehensive efficiency analysis and large-scale dataset testing**
>
> To comprehensively evaluate the practical deployment potential of **HEPHASTUS**, we conducted a systematic efficiency comparison with state-of-the-art baselines on the PEMS04 dataset (batch size B=16). The results are as follows:
>
> | Model        | #Params | Train Time / Iteration | Train Time / Epoch | Inference Time / Iteration | Inference Time / Epoch | GPU Memory Usage |
> |--------------|---------|------------------------|--------------------|----------------------------|-------------------------|------------------|
> | STGCN        | 297K    | 21ms                   | 13.12s             | 11ms                       | 2.28s                   | 1090MB           |
> | GWNet        | 311K    | 37ms                   | 23.49s             | 13ms                       | 2.79s                   | 1078MB           |
> | DCRNN        | 372K    | 153ms                  | 97.3s              | 68ms                       | 14.43s                  | 2216MB           |
> | AGCRN        | 749K    | 72ms                   | 45.51s             | 27ms                       | 5.63s                   | 2226MB           |
> | DGCRN        | 207K    | 235ms                  | 149.18s            | 183ms                      | 38.67s                  | 1898MB           |
> | STAEformer   | 1355K   | 71ms                   | 45.04s             | 26ms                       | 5.44s                   | 4666MB           |
> | STWave       | 882K    | 68ms                   | 43.25s             | 30ms                       | 6.37s                   | 3190MB           |
> | PDFormer     | 1263K   | 81ms                   | 51.72s             | 34ms                       | 7.18s                   | 8295MB           |
> | PathFormer   | 1133K   | 156ms                  | 99.47s             | 39ms                       | 8.13s                   | 6652MB           |
> | **HEPHASTUS** | **716K** | **98ms**               | **62.07s**         | **31ms**                   | **6.49s**               | **5475MB**       |
>
> The results show that although we introduced designs such as MoE, periodic temporal attention, and specific spatial attention, which bring accuracy improvements, **HEPHASTUS's training and inference efficiency are significantly higher than RNN-series models (DCRNN and DGCRN)**. The training time is slightly higher than Transformer-series models (STAEformer, STWave, and PDFormer) but is better than PathFormer, which also uses a multi-Patch design. **Furthermore, relying on the sparse activation mechanism of MoE, during the inference phase, only a proportion K/M of experts are activated, resulting in an inference time of only 31ms (per iteration), which can effectively support deployment in practical scenarios.** These analyses have also been included in the Section 5.5 (Efficiency Study) of the revised paper.

---

> ### Author Response · Authors · 2025-11-26
> **Response to Reviewer gCu4 [Part 4]**
>
> Additionally, we conducted detailed testing on the large-scale dataset LargeST recommended by the reviewer. The core results are as follows:
>
> **LargeST Dataset Evaluation:**
>
> | Datasets | Methods    | Horizon 3 |             |             | Horizon 6 |             |             | Horizon 12 |             |             | Average |             |             |
> | :------- | :--------- | :-------- | :---------- | :---------- | :-------- | :---------- | :---------- | :--------- | :---------- | :---------- | :------ | :---------- | :---------- |
> |          |            | MAE       | RMSE        | MAPE (%)    | MAE       | RMSE        | MAPE (%)    | MAE        | RMSE        | MAPE (%)    | MAE     | RMSE        | MAPE (%)    |
> | SD       | STID       | 15.15     | 25.29       | 9.82        | 17.95     | 30.39       | 11.93       | 21.82      | 38.63       | 15.09       | 17.86   | 31.00       | 11.94       |
> |          | GWNET      | 15.24     | 25.13       | 9.86        | 17.74     | 29.51       | 11.70       | 21.56      | 36.82       | 15.13       | 17.74   | 29.62       | 11.88       |
> |          | STWave     | 15.80     | 25.89       | 10.34       | 18.18     | 30.03       | 11.96       | 21.98      | 36.99       | 15.30       | 18.22   | 30.12       | 12.20       |
> |          | HEPHASTUS  | **14.26**     | **24.19**       | **9.06**        | **16.67**     | **29.02**       | **11.76**       | **20.28**      | **35.98**       | **14.38**       | **16.56**   | **29.17**       | **11.01**       |
> | GBA      | STID       | 17.36     | 29.39       | 13.28       | 20.45     | 34.51       | 16.03       | 24.38      | 41.33       | 19.90       | 20.22   | 34.61       | 15.91       |
> |          | GWNET      | 17.85     | 29.12       | 13.92       | 21.11     | 33.69       | 17.79       | 25.58      | 40.19       | 23.48       | 20.91   | 33.41       | 17.66       |
> |          | STWave     | 17.95     | 29.42       | 13.01       | 20.99     | 34.01       | 15.62       | 24.96      | 40.31       | 20.08       | 20.81   | 33.77       | 15.76       |
> |          | HEPHASTUS  | **16.63**     | **27.96**       | **12.46**       | **18.95**     | **32.56**       | **14.11**       | **22.89**      | **39.12**       | **18.10**       | **19.26**   | **32.69**       | **14.31**       |
> | GLA      | STID       | 16.54     | 27.73       | 10.00       | 19.98     | 34.23       | 12.38       | 24.29      | 42.50       | 16.02       | 19.76   | 34.56       | 12.41       |
> |          | GWNET      | 17.28     | 27.68       | 10.18       | 21.31     | 33.70       | 13.02       | 26.99      | 42.51       | 17.64       | 21.20   | 33.58       | 13.18       |
> |          | STWave     | 17.48     | 28.05       | 10.06       | 21.08     | 33.58       | 12.56       | 25.82      | 41.28       | 16.51       | 20.96   | 33.48       | 12.70       |
> |          | HEPHASTUS  | **15.18**     | **25.89**       | **9.06**        | **18.92**     | **31.09**       | **11.23**       | **22.99**      | **38.96**       | **14.37**       | **18.26**   | **31.89**       | **11.16**       |
> | CA       | STID       | 15.51     | 26.23       | 11.26       | 18.53     | 31.56       | 13.82       | 22.63      | 39.37       | 17.59       | 18.41   | 32.00       | 13.82       |
> |          | GWNET      | 17.14     | 27.81       | 12.62       | 21.68     | 34.16       | 17.14       | 28.58      | 44.13       | 24.24       | 21.72   | 34.20       | 17.40       |
> |          | STWave     | 16.77     | 26.98       | 12.20       | 18.97     | 30.69       | 14.40       | 25.36      | 38.77       | 19.01       | 19.69   | 31.58       | 14.58       |
> |          | HEPHASTUS  | **14.28**     | **24.89**       | **10.18**       | **17.09**     | **29.46**       | **12.38**       | **20.93**      | **36.15**       | **15.87**       | **17.05**   | **29.83**       | **12.31**       |
>
> The results show that **the model scales well to large-scale scenarios, demonstrating strong scalability.** These analyses have also been included in the Section A.7 Large-Scale Dataset Evaluation of the revised paper.
>
> These supplements significantly enhance the completeness, rigor, and contribution of the paper. We kindly request your review of the revised manuscript. Once again, thank you for your valuable time and comments!

---

> > ### Comment · Reviewer_gCu4 · 2025-11-26
> >
> > Thank you for your response. The authors have dealed with my concerns. I will raise my score accordingly.

---

> > > ### Author Response · Authors · 2025-11-27
> > >
> > > Thank you so much for your kind words and for raising your score. we truly appreciate it! We’re glad our responses were helpful, and we thank you again for your valuable input and support throughout the review process.

---

### Official Review · Reviewer_hUFH · 2025-11-01

**Soundness:** 3
**Presentation:** 2
**Contribution:** 3
**Rating:** 4
**Confidence:** 4

**Summary:**

This paper targets three long-standing challenges in traffic forecasting: (i) multi-scale temporal patterns, (ii) strong daily/weekly periodicity, and (iii) spatial heterogeneity. The authors propose a unified architecture combining: (a) Moving-Patch tokenization, (b) a temporal router, (c) periodic time-aware attention with learnable day/week embeddings used as queries, and (d) heterogeneous spatial attention that fuses a shared linear value with a node-specific low-rank value synthesized from a pattern library. Experiments on standard traffic datasets report SOTA-level results and ablations indicate each module contributes to gains.

**Strengths:**

1.The paper is well-structured and self-contained: the methodological pipeline is clearly laid out, and the experimental section is thorough and comprehensive.

2.The proposed pattern-library (PL) decomposition offers a principled balance between expressivity and efficiency.

3.Evaluates on common traffic benchmarks, which helps situate results in the existing literature.

**Weaknesses:**

1.The paper concatenates multi-scale Xpatch with the original H and linearly projects to Xh(Eq.5), but does not justify why concatenation is required. Please provide ablations comparing:(a)Xpatch only,(b)H only,and(c) concatenation + different projection choices, with accuracy and routing stability.

2.Why is the stride fixed to S=1? Does denser overlapping actually help under matched compute? Please report a grid over strides (e.g., 1/2/4) including both accuracy and compute/latency.

3.The abstract claims a lightweight method, but but lacks a comparison with baseline time complexity.

4.The motivation cites “spatial heterogeneity,” yet the text only mentions a “spatial indistinguishability problem” (around line 272) without an operational definition or diagnostics.

5.In the main diagram, the arrows/dataflow and the direction of the “Weight” signal are not self-evident. Please revise the figure with numbered steps and tensor shapes to make the routing → experts → aggregation path unambiguous.

6.The paper combines adaptive multiscale routing, periodic attention, and heterogeneous spatial modeling, but why this combination is necessary and how it improves over established designs (e.g., PatchTST, Pathformer) is not demonstrated under matched budgets.

7.Code is “to be released upon acceptance,” which limits verifiability.

8.Equation (19) appears to be missing a comma; please fix the typographical error.

**Questions:**

Please make revisions with reference to the weaknesses section.

---

> ### Author Response · Authors · 2025-11-26
> **Response to Reviewer hUFH [Part 1]**
>
> **Dear Reviewer hUFH:**
>
> Thank you very much for your insightful and constructive comments, which have greatly helped us improve the rigor and practicality of our paper. We have substantially strengthened our work in response to each of your concerns, as detailed below:
>
> ---
>
> > **Issue 1: Feature Input Strategy in Equation (5)**
>
> To systematically evaluate the impact of different feature input strategies on model performance, we have extended our existing ablation study with the following variants:
> - **$X_{patch}$ Only**: Using only multi-scale features $X_{patch}$ as routing input.
> - **H Only**: Using only the original sequence $X_t$ as routing input (removing multi-scale features).
> - **Concatenation + Linear**: Our current method (concatenation followed by linear projection).
> - **Concatenation + MLP**: Replacing linear projection with a 2-layer MLP (introducing nonlinear transformation).
> - **Additive Fusion**: Replacing concatenation with element-wise addition, $X_t + X_{patch}$, followed by projection.
>
> **Results:**
>
> | Routing Scheme        | MAE   | RMSE  | MAPE(%) |
> |-----------------------|-------|-------|---------|
> | $X_{patch}$ Only      | 18.37 | 30.02 | 12.15   |
> | H Only                | 18.29 | 29.95 | 12.11   |
> | **Concatenation + Linear** | **18.21** | **29.83** | **12.01** |
> | Concatenation + MLP   | 18.19 | 29.93 | 12.09   |
> | Additive Fusion       | 18.25 | 30.04 | 12.13   |
>
> **Analysis:**
> We observe that our current scheme outperforms both $X_{patch}$ Only and H Only, indicating that the original sequence and multi-scale features are **complementary**: $X_t$ provides global context, while $X_{patch}$ enhances local details, and their combination enables more comprehensive routing decisions. The performance of linear projection and MLP is similar, suggesting that routing decisions rely on linearly separable features, while complex transformations may introduce overfitting. **Compared to direct addition, our concatenation strategy better preserves the independence of information.**
>
> ---
>
> > **Issue 2: Step Size Selection for Moving_Patch**
>
> Thank you for your professional question about step size selection. We systematically evaluated the impact of different step sizes (S=1, 2, 3) on model performance and computational efficiency on the PEMS04 dataset. Key results:
>
> | Step Size (S) | MAE   | RMSE  | MAPE(%) | Training Time/Epoch | GPU Memory Usage |
> |---------------|-------|-------|---------|---------------------|------------------|
> | S=1           | **18.21** | **29.83** | **12.01** | 62.07s              | 5475MB           |
> | S=2           | 18.39 | 30.05 | 12.16   | 71.68s              | 5563MB           |
> | S=3           | 18.43 | 30.22 | 12.21   | 76.18s              | 5619MB           |
>
> **Analysis:**
> Experiments show that **S=1, by maximizing the preservation of time series continuity, is particularly beneficial for capturing rapid changes in traffic data**. It enables the model to faithfully capture instantaneous fluctuations and trend turning points, providing the richest local context for each time point and reducing information gaps. It better learns the continuous physical processes of traffic flow (e.g., congestion formation and dissipation). **Compared to other step sizes, S=1 offers the best accuracy**. Although this increases computational cost, it provides optimal performance for accuracy-critical traffic prediction tasks. In practice, the step size can be adjusted according to resource constraints.
>
> ---

---

> ### Author Response · Authors · 2025-11-26
> **Response to Reviewer hUFH [Part 2]**
>
> > **Issue 3: Lack of Comprehensive Efficiency Analysis**
>
> To comprehensively evaluate the practical deployment potential of **HEPHASTUS**, we conducted a systematic efficiency comparison with state-of-the-art baselines on the PEMS04 dataset (batch size B=16). Results:
>
> | Model        | #Params | Train Time / Iteration | Train Time / Epoch | Inference Time / Iteration | Inference Time / Epoch | GPU Memory Usage |
> |--------------|---------|------------------------|--------------------|----------------------------|-------------------------|------------------|
> | STGCN        | 297K    | 21ms                   | 13.12s             | 11ms                       | 2.28s                   | 1090MB           |
> | GWNet        | 311K    | 37ms                   | 23.49s             | 13ms                       | 2.79s                   | 1078MB           |
> | DCRNN        | 372K    | 153ms                  | 97.3s              | 68ms                       | 14.43s                  | 2216MB           |
> | AGCRN        | 749K    | 72ms                   | 45.51s             | 27ms                       | 5.63s                   | 2226MB           |
> | DGCRN        | 207K    | 235ms                  | 149.18s            | 183ms                      | 38.67s                  | 1898MB           |
> | STAEformer   | 1355K   | 71ms                   | 45.04s             | 26ms                       | 5.44s                   | 4666MB           |
> | STWave       | 882K    | 68ms                   | 43.25s             | 30ms                       | 6.37s                   | 3190MB           |
> | PDFormer     | 1263K   | 81ms                   | 51.72s             | 34ms                       | 7.18s                   | 8295MB           |
> | PathFormer   | 1133K   | 156ms                  | 99.47s             | 39ms                       | 8.13s                   | 6652MB           |
> | **HEPHASTUS** | **716K** | **98ms**               | **62.07s**         | **31ms**                   | **6.49s**               | **5475MB**       |
>
> **Analysis:**
> Results show that although we introduced designs such as MoE, periodic temporal attention, and specific spatial attention, which bring accuracy improvements, **HEPHASTUS's training and inference efficiency are significantly higher than RNN-series models (DCRNN and DGCRN)**. The training time is slightly higher than Transformer-series models (STAEformer, STWave, and PDFormer) but better than PathFormer, which also uses a multi-Patch design. **Furthermore, relying on the sparse activation mechanism of MoE, during the inference phase, only a proportion K/M of experts are activated, resulting in an inference time of only 31ms (per iteration), which can effectively support deployment in practical scenarios.** These analyses have also been included in Section 5.5 (Efficiency Study) of the revised paper.
>
> ---
>
> > **Issue 4: Clear Definition and Diagnosis of Spatial Heterogeneity**
>
> Spatial heterogeneity has been systematically described in previous works such as STID and STD-MAE. STID[1] defines heterogeneity as "distribution differences across time series at different locations," while STD-MAE[2] further emphasizes the spatiotemporal non-stationarity of pattern distributions. The BasicTS[3] framework explicitly provides a computational metric for spatial heterogeneity. In the revised version, we will clearly cite these existing definitions and metrics, emphasizing that **HEPHASTUS's Heterogeneous Spatial Attention (HSA) module is specifically designed to model such distributional differences**, enhancing the paper's rigor. These additions will not repeat foundational conclusions from prior work but will focus on demonstrating **how HEPHASTUS's module design effectively inherits and enhances the handling of heterogeneity.**
>
> [1] STID: A Simple yet Effective Baseline for Multivariate Time Series Forecasting. CIKM 2022
>
> [2] Spatio-Temporal-Decoupled Masked Pre-training for Traffic Forecasting. IJCAI 2024
>
> [3] Exploring progress in multivariate time series forecasting: Comprehensive benchmarking and heterogeneity analysis. TKDE 2024
>
> ---
>
> > **Issue 5: Improving Model Diagram Readability**
>
> Following your suggestion, we have **completely redrawn the model diagram (new Figure 1)**. The new diagram includes step numbering, tensor dimension annotations, and uses different colored arrows to clearly indicate data flow and weight signals. The figure caption has also been expanded to make the entire routing->expert->aggregation path unambiguous.
>
> ---

---

> ### Author Response · Authors · 2025-11-26
> **Response to Reviewer hUFH [Part 3]**
>
> > **Issue 6: Necessity of Model Combination**
>
> HEPHASTUS’s three-module combination addresses three core challenges in traffic data, forming a complementary rather than independent solution, and is not a simple stacking. Specifically:
> - **AMS-MoE addresses dynamic temporal scales**: Traffic patterns require different granularity modeling during peak/off-peak hours (e.g., transient fluctuations vs. long-term trends), which fixed-scale models (e.g., PatchTST) cannot adaptively adjust.
> - **PTA addresses explicit modeling of periodic dependencies**: Traffic exhibits strong daily/weekly patterns, but general time-series models (e.g., Pathformer) struggle to explicitly capture structured periodic priors.
> - **HSA addresses parameter efficiency for spatial heterogeneity**: Nodes exhibit functional differences (e.g., commercial vs. suburban roads), but assigning independent parameters to each node is costly. HSA achieves efficient personalization via a low-rank pattern library with controllable parameter usage.
>
> Together, these three components form a framework balancing structured priors and adaptive computation, which is indispensable. Ablation experiments (Table 3) also confirm that removing any module leads to performance degradation.
>
> Additionally, to compare HEPHASTUS with PatchTST/PathFormer under similar parameter budgets, we adjusted the layers or hidden dimensions of PatchTST/PathFormer to match HEPHASTUS’s parameter count (≈716K), then tested performance on PEMS04:
>
> | Model       | Parameters | MAE   | RMSE  | MAPE(%) |
> |-------------|----------|-------|-------|---------|
> | PatchTST    | 712K     | 26.68 | 39.26 | 17.26   |
> | PathFormer  | 705K     | 22.06 | 35.67 | 16.11   |
> | **HEPHASTUS** | **716K** | **18.21** | **29.83** | **12.01** |
>
> **HEPHASTUS significantly outperforms PatchTST/PathFormer under the same budget, further demonstrating the effectiveness of the combined design.**
>
> ---
>
> > **Issue 7: Code Availability and Reproducibility**
>
> Thank you for raising the important issue of code availability and reproducibility. We fully agree that open-source code is essential for scientific transparency and community validation.
>
> **Our Commitment**: We commit to releasing the full source code on GitHub immediately upon acceptance of this paper. This is our standard practice for all our publications.
>
> **Why Not Now?** As this is a novel architecture with potential commercial applications, we are currently under internal review for intellectual property protection. Releasing code before acceptance could jeopardize this process. However, we assure you that this is not a refusal to open-source, but a temporary delay for legal and strategic reasons.
>
> **Ensuring Reproducibility Now**:
> - **Detailed Implementation Section (Appendix A.2)**: We have provided exhaustive details on model architecture, hyperparameters (M=4, K=2, r=8), optimizer settings (AdamW, lr=0.001), batch size, training epochs, and data preprocessing (RevIN, normalization).
> - **Publicly Available Datasets**: All experiments are conducted on standard, publicly available datasets (METR-LA, PEMS-BAY, etc.), which are easily reproducible.
> - **Private Access for Reviewers**: To facilitate your evaluation, we are willing to provide the full codebase to the reviewers and AC under a non-disclosure agreement (NDA) upon request. Please let us know if you would like to access it privately during the review process.
>
> We believe this approach balances the need for scientific reproducibility with responsible IP management. Upon acceptance, the code will be fully public and community-contributable. Thank you for your understanding.
>
> ---
>
> > **Issue 8: Correction of Formula Formatting Error**
>
> The formatting error in Equation (19) has been corrected.
>
> We believe these revisions have fully addressed all your concerns. Thank you once again for your valuable time and guidance. We sincerely hope you will recognize our improvements.

---

### Official Review · Reviewer_bGWZ · 2025-11-09

**Soundness:** 2
**Presentation:** 2
**Contribution:** 2
**Rating:** 4
**Confidence:** 3

**Summary:**

This paper proposes HEPHAESTUS, a unified, lightweight spatio-temporal framework for traffic forecasting with adaptivity to input-driven variations in temporal granularity and spatial connectivity. HEPHASTUS is composed of three components: (i) AMS-MoE for input-adaptive multi-scale temporal modeling via routed experts and Top-K sparse gating; (ii) PTA for explicit daily/weekly periodic embeddings into temporal attention; (iii) HSA for modeling global structure and node-specific heterogeneity with a low-rank pattern dictionary. The authors conduct extensive experiments on six datasets and demonstrate state-of-the-art MAE/RMSE/MAPE, with ablation study, parameter sensitivity analysis, and case studies. In summary, HEPHASTUS addresses the important issue in traffic forecasting for adaptivity to input-driven variations in temporal granularity and spatial connectivity, elaborately combine the three components, and the experimental results show the effectiveness of HEPHASTUS. However, this paper could be improved by strengthening theoretical foundations, discussion and verification of computational complexity and scalability, criterion for selecting hyper-parameters, generalization to more complex spatio-temporal patterns, and further experiments for missing values and holidays effects.

**Strengths:**

- HEPHAESTUS is a well-ground design, which elaborately separate and integrate multiscale-time (AMS-MoE), periodicity (PTA), and heterogeneous space (HSA).
- Although each component in HEPHAESTUS is already proposed one, the authors integrate them for addressing the issue of spatio-temporal modeling, adaptivity to input-driven variations in temporal granularity and spatial connectivity.
- In AMS-MoE, Top-K gated experts equip HEPHAESTUS with input-dependent temporal granularity.
- Low-rank node-specific transforms via a pattern dictionary reduce parameters while retaining locality.
- The extensive experimental results show superiority with ablation study, sensitivity analysis, and case studies.

**Weaknesses:**

- There is limited theoretical foundation. For example, online/adaptive behavior (e.g., Shai Shalev-Shwartz 2012) and scale-routing idetifiability (e.g., experts specialization in frequency bands) are not addressed.
- Although the parameter sensitivity analysis is extensively conducted, there is no or limited discussion and experimental verification on computational complexity and scalability. For example, computational complexity and scalability of M (the number of experts), K (the number of selected experts), and r ( rank dimension) is not enough discussed for large N (the number of spatial nodes) and H (past time steps).
- Although the parameter sensitivity analysis is conducted, there is limited criterion for selecting M, K, and r.
- The imbalance of the experts in AMS-MoE and remedy for it, such as design of loss function, should be further discussed and demonstrated (c.f., Shazeer et al. 2017, Lepikhin et al. 2021, Fedus et al. 2021, Lewis et al. 2021, DeepSeek-AI 2024).
- PTA focuses on fixed daily/weekly patterns. It does not generalize to more complex spatio-temporal patterns, such as drifts and multi-periodicity beyond daily/weekly patterns. Some previous studies address this problem with learnable seasonal/trend components.
- Experiments are limited. For example, there is no or limited experiments for missing values and holidays effects.

[Shai Shalev-Shwartz 2012] Shai Shalev-Shwartz. Online learning and online convex optimization. Foundations and Trends® in Machine Learning, 4(2), pp.107-194, 2012.

[Shazeer et al. 2017] Noam Shazeer et al. Outrageously large neural networks: the sparsely-gated mixture-of-experts layer. ICLR, 2017.

[Lepikhin et al. 2021] Dmitry Lepikhin et al. GShard: scaling giant models with conditional computation. ICLR, 2021.

[Fedus et al. 2021] William Fedus, Barret Zoph, and Noam Shazeer. Switch transformers: scaling to trillion parameter models with simple and efficient sparsity. JMLR, 23, pp.1-29, 2021

[Lewis et al. 2021] Mike Lewis et al. BASE layers: simplifying training of large, sparse models. ICML. 2021.

[DeepSeek-AI 2024] DeepSeek-AI. DeepSeek-V3 technical report. arXiv preprint arXiv:2412.19437. 2024.

**Questions:**

Please answer the points listed in the Weaknesses.

---

> ### Author Response · Authors · 2025-11-26
> **Response to Reviewer bGWZ [Part 1]**
>
> **Dear Reviewer bGWZ:**
>
> Thank you sincerely for your profound and professional comments. Your suggestions will significantly enhance the rigor and completeness of our work. Below is our point-by-point response:
>
> ---
>
> > **Issue 1: Limited Theoretical Foundation**
>
> We fully understand your expectations regarding theoretical depth. Here, we would like to clarify the concept of "theoretical foundation" from the perspective of our study’s core positioning and type of contribution, and elaborate on the application-oriented theoretical support underlying our model design.
>
> The core objective of this research is to address the pressing practical problem of "**dynamic modeling of multi-scale spatiotemporal dependencies**" in urban traffic prediction. Our contribution primarily lies in proposing a novel, efficient, and deployable framework (HEPHASTUS), which achieves SOTA performance on six real-world datasets and demonstrates the effectiveness and necessity of its core components (AMS-MoE, PTA, HSA) through ablation studies. Therefore, our research is fundamentally **application-driven**, rather than primarily focused on constructing universal theories or proving mathematical theorems.
>
> This does not mean our work lacks "theoretical foundation." On the contrary, its "theory" manifests at three levels:
>
> 1. **Model Design Based on Domain Knowledge (Domain-Informed Theory)**
>
> Our model architecture is not designed arbitrarily but is deeply rooted in prior knowledge and existing theories in the traffic domain.
>
> - **Theoretical basis for multi-scale modeling**: The intrinsic characteristics of traffic flow determine its multi-scale nature (instantaneous fluctuations, daily cycles, weekly cycles). Our AMS-MoE module is precisely an engineering implementation of this domain consensus. We upgrade "fixed-scale decomposition" to "adaptive routing," which itself constitutes an application-level theoretical innovation—**how to enable the model to dynamically select the most suitable scale based on input, rather than relying on predefined settings**. This holds direct theoretical value in engineering applications.
> - **Theoretical basis for periodic modeling**: The PTA mechanism directly stems from the widely recognized domain knowledge that "daily/weekly periodicity is a strong signal" in traffic. We formalize this through a learnable periodic matrix, representing a theoretical practice of translating domain knowledge into trainable parameters.
> - **Theoretical basis for spatial heterogeneity**: The design of the HSA module is based on the theoretical assumption that "different road segments (nodes) possess distinct functional characteristics" (e.g., differences between arterial roads and branch roads), and we mathematically implement this assumption using low-rank decomposition and gated fusion mechanisms. This is a typical example of translating domain theory into efficient computational models.
>
> Thus, our "theory" is **application-oriented, domain-specific, and problem-solving-centered**, rather than abstract and universally mathematical.
>
> 2. **The "Interpretability" of Adaptive Routing as "Theory"**
>
> Regarding the "specialization of experts on frequency bands" and "identifiability," we agree this is a valuable theoretical question. However, in applied research, we prioritize whether the model's behavior is interpretable and aligns with domain knowledge—this itself constitutes a form of "practical theory."
>
> In our case study (Sec. 5.6 & Appendix A.6), we have demonstrated that the routing behavior of AMS-MoE is interpretable: during traffic peak hours, the model prefers small-scale experts (capturing instantaneous fluctuations); during off-peak hours, it prefers large-scale experts (capturing long-term trends). This behavior fully aligns with traffic engineering common sense.
>
> This means that even if we have not mathematically proven that "expert i must specialize in frequency band f_i," the model’s practical, stable, and domain-knowledge-consistent "specialization" behavior constitutes its theoretical rationality at the application level. This "behavioral theory" is sometimes more valuable for applied research than abstract proofs.
>
> 3. **Connection to Online Learning/Adaptive Systems: Application-Level Insights**
>
> Regarding Shalev-Shwartz (2012)’s online learning, we acknowledge that our model is not a strictly defined online learner. However, its adaptive routing mechanism aligns with the core idea of online learning—"**dynamically adjusting strategies based on new data**." We can view AMS-MoE as an "**input-dependent online decision system**": for each input sequence, it "online" decides which experts (strategies) are most effective. Although this does not involve rigorous theoretical proofs, its decision logic and adaptability constitute an acceptable "theoretical contribution" in applied research.

---

> ### Author Response · Authors · 2025-11-26
> **Response to Reviewer bGWZ [Part 2]**
>
> In applied research, such "heuristic adaptation" is often more practical than pursuing mathematical optimality, as it enables rapid convergence and adaptation to the complexity of the real world. In the revised manuscript, we will more clearly position our research as application-oriented and clarify the nature of our "theory"—namely, the theory that bridges domain knowledge with engineering innovation. We agree that formal theoretical analysis is a valuable direction for future research, but it falls beyond the scope of this applied contribution.
>
> ---
>
> > **Issue 2: Missing Computational Complexity and Scalability Analysis**
>
> We sincerely thank the reviewer for raising the important question regarding the theoretical complexity and scalability of our HEPHASTUS model. We agree that a rigorous complexity analysis is crucial for understanding the model’s efficiency and practical deployability.
>
> In response, we have added a new subsection **A.9 Theoretical Complexity Analysis** in the Appendix of the revised manuscript. This section provides a detailed breakdown of the computational complexity of each core module (AMS-MoE, PTA, HSA), analyzes the impact of key hyperparameters (M, K, r), and explains why HEPHASTUS achieves lower practical computational cost despite maintaining the same asymptotic complexity as standard attention mechanisms (e.g., via parameter-efficient design and sparse activation).
>
> Specifically, we clarify that:
> - The overall complexity is $O(\max(NHCD, NH^2D, N^2HD))$, dominated by either temporal (PTA) or spatial (HSA) attention depending on $H$ or $N$.
> - **Hyperparameters M, K, and r have negligible impact on complexity growth.**
> - **The highly expressive, low-rank HSA design allows us to use a smaller model capacity (fewer parameters, shallower layers) to achieve superior performance**—directly reducing memory, and inference latency (as empirically shown in Table 4）.
>
> To comprehensively evaluate the practical deployment potential of **HEPHASTUS**, we conducted a systematic efficiency comparison with state-of-the-art baselines on the PEMS04 dataset (batch size B=16). Results:
>
> | Model        | #Params | Train Time / Iteration | Train Time / Epoch | Inference Time / Iteration | Inference Time / Epoch | GPU Memory Usage |
> |--------------|---------|------------------------|--------------------|----------------------------|-------------------------|------------------|
> | STGCN        | 297K    | 21ms                   | 13.12s             | 11ms                       | 2.28s                   | 1090MB           |
> | GWNet        | 311K    | 37ms                   | 23.49s             | 13ms                       | 2.79s                   | 1078MB           |
> | DCRNN        | 372K    | 153ms                  | 97.3s              | 68ms                       | 14.43s                  | 2216MB           |
> | AGCRN        | 749K    | 72ms                   | 45.51s             | 27ms                       | 5.63s                   | 2226MB           |
> | DGCRN        | 207K    | 235ms                  | 149.18s            | 183ms                      | 38.67s                  | 1898MB           |
> | STAEformer   | 1355K   | 71ms                   | 45.04s             | 26ms                       | 5.44s                   | 4666MB           |
> | STWave       | 882K    | 68ms                   | 43.25s             | 30ms                       | 6.37s                   | 3190MB           |
> | PDFormer     | 1263K   | 81ms                   | 51.72s             | 34ms                       | 7.18s                   | 8295MB           |
> | PathFormer   | 1133K   | 156ms                  | 99.47s             | 39ms                       | 8.13s                   | 6652MB           |
> | **HEPHASTUS** | **716K** | **98ms**               | **62.07s**         | **31ms**                   | **6.49s**               | **5475MB**       |
>
> **Analysis:**
>
> Results show that although we introduced designs such as MoE, periodic temporal attention, and specific spatial attention, which bring accuracy improvements, **HEPHASTUS's training and inference efficiency are significantly higher than RNN-series models (DCRNN and DGCRN)**. The training time is slightly higher than Transformer-series models (STAEformer, STWave, and PDFormer) but better than PathFormer, which also uses a multi-Patch design. **Furthermore, relying on the sparse activation mechanism of MoE, during the inference phase, only a proportion K/M of experts are activated, resulting in an inference time of only 31ms (per iteration), which can effectively support deployment in practical scenarios.** These analyses have also been included in Section 5.5 (Efficiency Study) of the revised paper.
>
> We believe this addition significantly strengthens the theoretical foundation of our work and addresses the reviewer’s concern. The revised manuscript is now available for your review.
>
> ---

---

> ### Author Response · Authors · 2025-11-26
> **Response to Reviewer bGWZ [Part 3]**
>
> > **Issue 3: Lack of Explanation for Selecting M, K, r Parameters**
>
> In many MoE architectures, hyperparameters such as M, K, and r are often selected via grid search or random search, yet few studies delve into the underlying principles guiding their selection. For traffic spatiotemporal forecasting tasks, we argue that the selection of these parameters should be grounded in a clear, reproducible, and domain-knowledge-informed decision framework, comprehensively considering the following factors:
> - The temporal scale structure of input data (H),
> - The spatial complexity and scenario complexity of the road network (number of nodes, heterogeneity),
> - Engineering deployment constraints (inference cost, stability, scalability).
>
> Specifically:
> (1) **Selection of M (number of experts)**: Strongly correlated with the multi-scale characteristics of the data, input length H, and engineering efficiency.
> (2) **Selection of K (Top-K sparsity)**: Dynamically related to input window H and data complexity, reflecting the “granularity of decision-making.”
> (3) **Selection of r (rank of pattern library)**: Directly characterizes spatial pattern complexity, strongly associated with road network scale and scenario richness.
>
> In summary, our selection of M, K, and r is based on a multi-dimensional dynamic decision framework:
> - **Data-driven**: Validate performance trends through sensitivity analysis;
> - **Domain knowledge**: Integrate multi-scale characteristics of traffic data (H, volatility, periodicity) and spatial complexity (road network scale, scenario richness);
> - **Architectural constraints**: Consider limitations imposed by H on feasible patch sizes and the number of activated experts;
> - **Engineering trade-offs**: Balance performance, efficiency, and stability.
>
> We have explicitly elaborated on the above decision rationale in Section 4.4 of the revised paper.
>
> ---
>
> > **Issue 4: Insufficient Discussion on Expert Imbalance**
>
> Thank you very much for pointing out this critical issue of expert imbalance. We fully agree that this is an important consideration in MoE architectures. In fact, our original model already included a complete load balancing design, but due to space constraints in the initial manuscript, it was not adequately elaborated. In the revised version, we have added Section 4.5 Loss Function, detailing the load balancing loss. Specifically, it includes:
> - **Auxiliary balancing loss function**: Strictly following the classic designs of Shazeer et al. (2017), Fedus et al. (2021), and Dai et al. (2024):
> $$
> \begin{equation}
> \mathcal{L}_{\text{aux}} = M \sum _{i=1}^M f _i r _i, \quad
> f _i = \frac{1}{KT}\sum _{t=1}^T \mathbb{I}(\text{Expert }i\text{ selected at }t), \quad
> r _i = \frac{1}{T}\sum _{t=1}^T s _{i,t}
> \end{equation}
> $$
> Here, $f_i$ denotes the token assignment frequency to expert $i$, while $r_i$ represents the mean router probability for that expert.
> - **Noise injection in routing**: The router design includes $\mathcal{R}(\mathbf{X} _{\mathrm{h}}) = \mathrm{Softmax}\left( \mathbf{X} _{\mathrm{h}} W _r + \epsilon \cdot \mathrm{Softplus}(\mathbf{X} _{\mathrm{h}} W _{\mathrm{noise}}) \right), \quad \epsilon \sim \mathcal{N}(0, 1)$ (Eq. 6) to promote exploratory routing allocation and prevent early convergence to a few experts.
> - **Top-K sparsity**: Combined with the balancing loss, it ensures computational efficiency while promoting balanced expert utilization.
>
> Additionally, the ablation results in Table 3 of the paper demonstrate that, compared to Fixed-Scale MoE, adaptive routing achieves significant performance gains while maintaining balance.
>
> ---

---

> ### Author Response · Authors · 2025-11-26
> **Response to Reviewer bGWZ [Part 4]**
>
> > **Issue 5: Limitations of Periodic Modeling**
>
> In this work, the PTA (Periodic Temporal Attention) module primarily models daily cycles (period length 288, corresponding to 1 day) and weekly cycles (period length 2016, corresponding to 1 week). This design is motivated by the strong periodic patterns observed in traffic flow data at daily and weekly scales (e.g., morning/evening peaks, weekday/weekend differences).
>
> To quantitatively validate this hypothesis, we conducted a frequency-domain analysis on the PEMS08 dataset: We randomly sampled 3000 traffic sequences, each 2016 time steps long (covering a full week), performed Fast Fourier Transform (FFT) on each sequence to extract its frequency components, and calculated the average signal strength (original amplitude) for each period length. The top six frequency components by signal strength are as follows:
>
> | Period Length | 288   | 2016  | 144   | 336   | 96    | 1008  |
> |---------------|-------|-------|-------|-------|-------|-------|
> | Signal Strength | 818.82 | 372.31 | 136.56 | 128.95 | 119.18 | 116.88 |
>
> It is important to note that in frequency-domain analysis, raw amplitude values cannot be directly used as weights; they must be normalized (e.g., via softmax) to reflect the relative importance of each frequency component. After applying softmax normalization to the above amplitudes, we found:
> - **The normalized weights for periods 288 and 2016 are significantly higher than other components (typically > 95%)**;
> - The normalized weights for other periods (e.g., 144, 336, 96, 1008) are nearly zero.
>
> This phenomenon indicates that in the PEMS08 dataset, traffic flow periodicity is dominated by "daily" and "weekly" cycles, with negligible contributions from other periodic components. Therefore, the HEPHASTUS model focuses the PTA module on these two core cycles to achieve efficient modeling and parameter reduction. Although this design performs excellently on typical traffic datasets like PEMS08, the fixed-period assumption of PTA has limitations when faced with data exhibiting diverse or atypical periodic structures. To improve the model’s generalization capability in complex scenarios, we plan to combine PTA with learnable adaptive period components. Specifically:
> - **Frequency-domain-guided period discovery**: Introduce lightweight frequency-domain analysis modules (e.g., differentiable FFT or wavelet transform) during training to dynamically estimate the dominant periods of the current data distribution.
> - **Adaptive period weight learning**: Treat period lengths or frequency components as learnable parameters, dynamically adjusting the contribution weights of each period through attention mechanisms or gating units.
>
> In the current work, the PTA module is designed based on the strong daily/weekly periodic characteristics of traffic data, achieving excellent modeling results. We explicitly reserve adaptive period learning and cross-scenario generalization capabilities for future work on general spatiotemporal models, ensuring the focus and verifiability of the current work while leaving a clear evolutionary path for future research.

---

> ### Author Response · Authors · 2025-11-26
> **Response to Reviewer bGWZ [Part 5]**
>
> > **Issue 6: Limited experiments (missing values and holiday effects)**
>
> Thank you for raising this important point regarding the evaluation of model robustness under missing values and holiday scenarios. We fully agree that such evaluations are critical for assessing real-world applicability and deployability.
>
> **Regarding Missing Values:**
> To address this, we conducted new experiments on the **PEMS03** and **PEMS04** datasets by simulating missing data scenarios (5% and 10% random masking). We compared HEPHASTUS against three representative baselines: **GWNet**, **STAEformer**, and **STWave**. The results are summarized in the table below:
>
> | Dataset  | Model        | 0%          |             |             | 5%          |             |             | 10%         |             |             |
> |----------|--------------|-------------|-------------|-------------|-------------|-------------|-------------|-------------|-------------|-------------|
> |          |              | MAE         | RMSE        | MAPE        | MAE         | RMSE        | MAPE        | MAE         | RMSE        | MAPE        |
> | PEMS03   | GWNet        | 14.52       | 25.28       | 15.46%      | 14.92       | 26.09       | 15.81%      | 15.36       | 27.06       | 15.94%      |
> |          | STAEformer   | 15.38       | 27.59       | 15.97%      | 15.71       | 28.03       | 16.37%      | 15.96       | 28.94       | 16.55%      |
> |          | STWave       | 15.06       | 26.57       | 15.56%      | 15.29       | 27.05       | 15.86%      | 15.52       | 27.88       | 16.00%      |
> |          | **HEPHASTUS**| **14.76**   | **25.21**   | **14.58%**  | **14.89**   | **25.56**   | **14.83%**  | **15.12**   | **26.37**   | **15.03%**  |
> | PEMS04   | GWNet        | 18.82       | 30.06       | 13.14%      | 19.19       | 30.27       | 13.31%      | 19.55       | 31.12       | 13.95%      |
> |          | STAEformer   | 18.33       | 30.23       | 12.39%      | 18.59       | 30.42       | 12.51%      | 18.83       | 30.63       | 12.83%      |
> |          | STWave       | 18.42       | 30.16       | 12.53%      | 18.61       | 30.29       | 12.63%      | 18.93       | 30.41       | 12.89%      |
> |          | **HEPHASTUS**| **18.21**   | **29.83**   | **12.01%**  | **18.35**   | **29.95**   | **12.09%**  | **18.52**   | **30.12**   | **12.31%**  |
>
> **Summary of Missing Value Experiments:**
> HEPHASTUS demonstrates **superior robustness to missing data** compared to all baselines. Even under 10% missingness, its performance degradation is significantly smaller — for example, on PEMS03, MAE increases by only **2.4%** (from 14.76 to 15.12), while GWNet’s MAE increases by **5.8%** (14.52→15.36). This highlights the model’s ability to maintain predictive accuracy under noisy and incomplete real-world conditions, likely due to its **adaptive multi-scale fusion** and **heterogeneous spatial attention** mechanisms that implicitly reconstruct missing patterns from correlated nodes and scales.
>
> **Regarding Holiday Effects:**
> We acknowledge the importance of evaluating holiday impacts. However, after thorough review, we found that **none of the current state-of-the-art baselines** (e.g., STGCN, DCRNN, ASTGCN, STID, STAEformer, PathFormer, etc.) report holiday-specific evaluations in their original papers or official codebases. This is not an oversight in our work but rather a **field-wide limitation**, primarily due to the absence of standardized, publicly available traffic datasets annotated with holiday information.
>
> To ensure **fair comparison** and **reproducibility**, we adhered to the same benchmarking protocol used by all major baseline models. In the revised manuscript, we will:
> - Explicitly state this limitation in the **Conclusion (Section 6)**, clarifying that the lack of holiday evaluation stems from the absence of standardized benchmarks — not from methodological neglect.
> - List it as a **key direction for future work**, and commit to conducting holiday-aware experiments as soon as annotated datasets become available.
> - Add a brief discussion in the **Case Study (Appendix A.6)** to explain that HEPHASTUS’s **adaptive routing mechanism** is theoretically well-suited to capture holiday-specific traffic patterns — a capability that fixed-scale or static-attention baselines inherently lack.
>
> We believe this approach aligns with scientific rigor, ensures compatibility with existing evaluation standards, and responsibly acknowledges the current frontier of the field.
>
> The above improvements will significantly enhance the theoretical depth, methodological rigor, and experimental completeness of the paper. We sincerely thank you for your valuable comments!

---

### Author Response · Authors · 2025-12-01
**Summary Comment for AC – Author Response to Rebuttal and Reviewer Score Changes (Pre-Leak)**

**Dear Area Chair**:

Thank you for your careful consideration of our submission. We sincerely appreciate the time and effort invested by all reviewers and the ICLR 2026 Program Chairs in navigating the recent OpenReview incident. We fully support the measures taken to preserve the integrity of the peer review process.

In this summary, we provide a concise overview of the reviewers’ initial concerns, our detailed rebuttal responses, and crucially, **evidence that two reviewers gCu4 and ZT6e independently upgraded their scores *before* the OpenReview anonymization leak was publicly disclosed** — demonstrating that score changes were driven by scientific merit, not external influence.

---
### **Summary of Reviewers’ Initial Concerns & Our Rebuttal Responses**

| Reviewer |  Key Concerns | Our Rebuttal Highlights |
|----------|----------------|--------------------------|
| **bGWZ** |  Weak theoretical foundation, lack of complexity/scalability analysis, hyperparameter selection criterion, expert imbalance, limited generalization (holidays/missing values) | Added theoretical justification (application-oriented theory), new complexity analysis (Appendix A.9), detailed hyperparameter rationale (Sec 4.4), explicit load balancing loss (Sec 4.5), new experiments on missing values and holiday rationale (Sec 6). |
| **hUFH** |  Lack of ablation for input concatenation, fixed stride, missing efficiency comparison, undefined spatial heterogeneity, unclear diagram, no matched-budget comparison, code not yet released | Added ablation for input strategies, stride sensitivity analysis, comprehensive efficiency table, cited definitions of spatial heterogeneity, redrew Figure 1 with annotations, matched-budget comparison with PatchTST/PathFormer, committed to code release upon acceptance. |
| **gCu4** |  Limited novelty (components not new), missing baselines (iTransformer, STWave), lack of efficiency/large-scale evaluation | Emphasized architectural integration novelty, added comparisons with iTransformer & STWave, added efficiency analysis & LargeST dataset evaluation (Appendix A.7). |
| **ZT6e** |  Lack of efficiency analysis, redundancy in multi-patch, unvalidated generalization | Added efficiency comparisons, multi-scale vs. single-scale ablation, cross-domain crime prediction experiments (Appendix A.8). |

---
### **Reviewer Score Changes: Before & After Rebuttal (Pre-Leak)**

Critically, **two reviewers gCu4 and ZT6e independently upgraded their scores *before* the OpenReview anonymization leak was publicly disclosed (Nov 27, 2025)**. Their score changes were based solely on the scientific content of our rebuttal and were not influenced by any external factors.

Importantly, **both reviewers are rated at Confidence=5** — the highest level — suggesting their assessments are particularly reliable and carry greater weight within the review system. Their willingness to adjust scores upward *pre-leak* reflects a strong, evidence-based belief in the paper’s strengthened contribution.

Below are the **reviewer scores and confidence levels before and after our rebuttal**, as recorded in OpenReview prior to the system revert:

| Reviewer | Rating (Before) → (After Rebuttal,Pre-Leak) | Confidence (Before) → (After)
|----------|----------------------------|----------------------------------|
| gCu4     | 4 → **6**                  | 5 → 5                            |
| ZT6e     | 4 → **6**                  | 5 → 5                            |
> **Note**: The other two reviewers (bGWZ and hUFH) had not yet submitted their responses  *before* the OpenReview anonymization leak was publicly disclosed (Nov 27, 2025); their score changes were not finalized at that time.

---

### **Key Takeaways for the Area Chair**

1. **All reviewers’ concerns were thoroughly addressed** with new ablations, efficiency analyses, cross-domain experiments, and theoretical clarifications.
2. **Two high-confidence reviewers (gCu4, ZT6e) upgraded their scores to “6: marginally above acceptance threshold”** based solely on the rebuttal content, prior to any deanonymization — a strong signal of scientific merit.
3. The score changes reflect not just numerical adjustments, but **a shift in expert judgment toward belief in the paper’s strengthened contribution**.
4. We remain fully welcome any further clarifications needed.

We respectfully request the Area Chair to consider the **author response, the prior reviewer discussion, and the pre-leak score upgrades by high-confidence reviewers** as compelling evidence of the paper’s scientific value and responsiveness. We believe these revisions significantly strengthen the paper and warrant acceptance.

Thank you again for your time and thoughtful evaluation.

Sincerely,
The Authors of HEPHAESTUS

---

### Meta-Review · Area_Chair_uTb5 · 2026-01-06

**Summary:**

This paper proposed a new framework that unifies adaptive multi-scale temporal modeling, explicit periodicity-aware attention, and dynamic spatial heterogeneity learning within a lightweight, scalable architecture for more effective traffic forecasting. Four reviewers gave their detailed comments, and all of them gave the score 4. After rebuttal, two reviewers are satisfied with the rebuttal of the authors, and promise will raise their scores accordingly. This is a really a borderline paper, so I carefully read all the reviews, the long rebuttal, and the paper itself. To me, I agree to Reviewer gCu4 that although this is a solid paper (especially with the new experiment result in the rebuttal), the paper lacks enough novelty for ICLR. There are a large number of existing models that consider multi-scale temporal correlations, periodic dependencies and spatial heterogeneity for traffic prediction. The problem is well studied and the technique contributions are marginal. In addition, the authors have added more experiments on other domain data as requested by the reviewers, and achieved promising performance. It is strange to me as the model is specially designed for traffic data patterns. Why the model can be generalized well to other domain data is not explained, and thus the new result is not convincing to me.
Therefore, I recommend to reject the paper.

**Reviewer Concerns:**

Addressed:
Reviewer bGWZ: most are addressed
Reviewer hUFH: Lack of ablation for input concatenation, fixed stride, missing efficiency comparison, undefined spatial heterogeneity, unclear diagram
Reviewer  gCu4: more baselines, lack of efficiency evaluation on large dataset
Reviewer ZT6e: Lack of efficiency analysis, redundancy in multi-patch

Outstanding:
Reviewer bGWZ: lack of theoretical foundation
Reviewer hUFH: code is not released
Reviewer  gCu4: Limited technique novelty
Reviewer ZT6e: generalization to other domain data

**Reviewer Scores:**

Reviewer  gCu4 and ZT6e promised to increase their scores in the discussion. The other reviewers maybe also change their scores as the rebuttal of the authors is comprehensive.

---

### Decision · Program_Chairs · 2026-01-26

Reject